

# Assimilation of SMOS soil moisture into a distributed hydrological model and impacts on the water cycle variables over the Ouémé catchment in Benin

Delphine J. Leroux[1,2], Thierry Pellarin[1], Théo Vischel[1], Jean-Martial Cohard[1], Tania Gascon[1], François Gibon[1], Arnaud Mialon[2], Sylvie Galle[1], and Christophe Peugeot[3]

[1]LTHE, Laboratoire d'étude des Transferts en Hydrologie et Environnement, Grenoble, France.
[2]CESBIO, Centre d'Etudes Spatiales de la Biosphère, Toulouse, France.
[3]HydroSciences, Montpellier, France.

*Correspondence to:* Delphine J. Leroux (delphine.j.leroux@gmail.com)

**Abstract.** The impact of the assimilation of surface soil moisture on the simulations of the physically based hydrological model DHSVM (Distributed Hydrology Soil Vegetation Model) is investigated in this paper for a 12,000 km$^2$ catchment located in Benin, West Africa. Thanks to a large number of rain gauges spread all over the entire basin, reference simulations are performed from one year of calibration (in 2010) and two years of evaluation (2011 and 2012) based on in situ measurements

of streamflow at the outlet and local observations of soil moisture at different soil depths and evapotranspiration. In a second step, several satellite products (PERSIANN, TRMM-3B42RT, and CMORPH) are used instead of in situ precipitation measurements. These products bring too much water (especially PERSIANN and CMORPH), sometimes not at the correct time of the year, which has a large impact on various hydrological variables. In order to correct for the wrong amount of input water brought by the satellite precipitation products, the SMOS satellite soil moisture observations are assimilated in the hydrological

model. An optimal interpolation is implemented here using an influence radius in order to replicate the field of view of the SMOS instrument. The assimilation of SMOS data shows a positive impact on the soil moisture at different depths (5 cm, 40 cm, and 80 cm defined in the model), with a decrease of the bias compared to the in situ measurements. Streamflow is also positively impacted with a large improvement of the Nash efficiency coefficient after assimilation (from negative to positive for PERSIANN and CMORPH). Finally, the temporal evolution of the water table depth is also greatly improved (from 0.1-0.3

to 0.8-0.9 for PERSIANN and CMORPH). This work shows that the use of satellite precipitation products into a hydrological model can lead to large errors that can be reduced by assimilating satellite soil moisture, which has a positive impact on the estimation of hydrological variables at deeper layers and at other stages of the water cycle.

## 1  Introduction

Water is one of the most valuable resources and has an undeniable influence on every aspect of life on Earth. It is present in

different forms and different quantities. While most of the water can be found in the ocean, fresh water only represents 2.5% available in the form of ice or snow, vapor or liquid. Hydrological models have been developed for a better understanding



of the hydrologic processes governing the water fluxes from the atmosphere to the deep ground, that are most of the time coupled with energy fluxes. With a better understanding comes a better assessment of water resources and better forecasts as well. Precipitation is considered as the main input source of water and is then usually used as a crucial forcing of hydrological models.

The soil is at the interface between the water and the energy cycles, or between the deep soil and the atmosphere. The soil water content contributes to the water flow (at the surface or sub-surface) and to the infiltration along with the soil properties and the intensity of the precipitations. It is thus essential to represent correctly this amount of water contained in the soil in the hydrological models. Surface soil moisture, which can be measured and observed from space, is a good indicator of the partitioning of the precipitation into surface runoff and infiltration (water cycle), and also of the partitioning of the incoming solar and atmospheric radiations into latent, sensible, and ground heat fluxes (energy balance).

Ground measurements of soil moisture are broadly used to monitor the hydrological cycle of a specific region. As all in situ stations, the soil moisture probes need to be maintained and are most of the time installed for a limited amount of time. Moreover, the spatial heterogeneity representation of the catchment is limited by the number of probes available for the experiment. Soil moisture monitoring from space has thus been developed for a larger/wider coverage in space and assures continuity in time as long as the space mission is still operating. These two types are very complementary with in situ stations being able to directly measure soil moisture profiles at different depths and also used for satellite soil moisture validation.

SMOS (Soil Moisture and Ocean Salinity, Kerr et al. (2010)) is the first ESA's Earth Explorer mission specifically designed for soil moisture monitoring over land. It was launched in November 2009 and has been providing since global maps of soil moisture every three days using its interferometric radiometer operating at L-band (1.4 GHz, most sensitive frequency for soil moisture monitoring, Kerr et al. (2001)). An inversion algorithm based on a radiative transfer model (Wigneron et al., 2007) generates soil moisture products on a 25 km grid with an accuracy of 0.04 $m^3/m^3$. More recently, the SMAP (Soil Moisture Active-Passive, Entekhabi et al. (2010)) NASA's mission, launched in January 2015, has embedded one radiometer and one radar, both observing the Earth at L-band. The novelty of this mission is to use the finer resolution of the radar coupled to the sensitivity of the radiometer to retrieve the soil moisture with a better resolution (gridded on a 9 km grid) with the same accuracy. Unfortunately, due to a power failure of the radar instrument a few weeks after launch, only the soil moisture derived from the radiometer is available on a 36 km grid. The availability of these two remote sensing observations has been enhancing the soil moisture and the water cycle monitoring at the global scale.

In order to take advantage of these dedicated space missions, the hydrological model simulations can be merged with available observations through data assimilation. This technique has already been widely used by weather forecast models at regional and global scales, using the remote sensing observations and ground measurements to adjust their forecasts in time.

Various variables related to the water cycle have been used for assimilation. GRACE terrestrial water storage products (Gravity Recovery and Climate Experiment, Rowlands et al. (2005)) have also been used for assimilation. The Ensemble Kalman Smoother (Ensemble Kalman filter using a time window) has been used to account for the monthly average aspect of the GRACE product such as over the Mississipi basin in the US (Zaitchik et al., 2008), nine large basins in Europe (Li et al., 2012a), and the Mackenzie river basin in Canada (Forman et al., 2012). Despite the coarse resolution of the GRACE





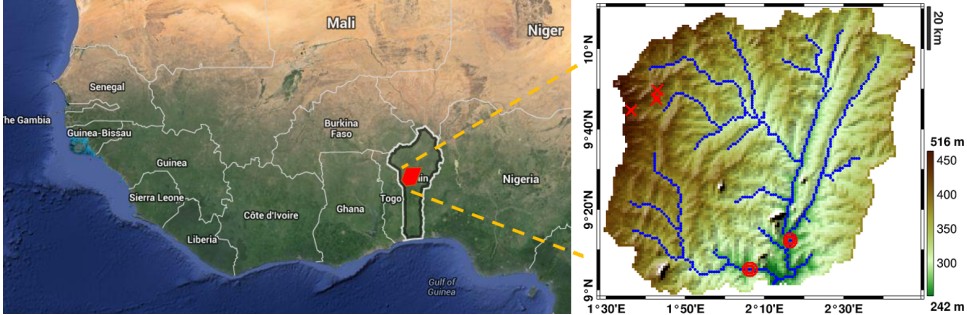

**Figure 1.** The Ouémé catchment is located in Benin, West Africa. On the right panel are indicated the location of three soil moisture stations in the North-Western part (red crosses), and two streamflow sensors installed in the Southern part (red circles, the outlet total streamflow being the sum of the two stations measurements).

observations (product available at a 300 km spatial resolution), the assimilation improves the estimation of the groundwater volume (Zaitchik et al., 2008) and thus improves drought monitoring by also having a positive impact on the runoff simulation (Li et al., 2012a).

Many previous studies have already used soil moisture assimilation for hydrological purposes at catchment scale. With the availability of more than thirty five years of soil moisture at the global derived from a series of satellites (SMMR, SSM/I, TRMM-TMI, AMSR-E, ASCAT, Windsat, Liu et al. (2011, 2012); Wagner et al. (2012)), the soil moisture CCI (Climate Change Initiative by ESA, Hollmann et al. (2013)) has been assimilated in many models for hydrological purposes such as the streamflow simulation (Pauwels et al., 2001, 2002), flood events via runoff simulation (Brocca et al., 2010, 2012), drought prediction (Kumar et al., 2014), root zone soil moisture simulations (Draper et al., 2012; Renzullo et al., 2014) for a better prediction of agricultural yields (Chakrabarti et al., 2014). Han et al. (2012) voluntarily degraded the precipitation input and showed that soil moisture, water table depth and evapotranspiration simulations could be improved by assimilating surface soil moisture. As most of the soil moisture assimilation studies, Ridler et al. (2014) have found that it improves the distribution of the soil moisture simulations but has little impact on the streamflow estimation.

At the catchment scale, additional issues arise when assimilating satellite observations into models. The first issue is the scale difference between the observations and the simulations. If we refer to passive microwave observations (SMOS and SMAP), the spatial resolution of the observation is around 40 km whereas a hydrological model can be run at a sub-kilometer to a few kilometer resolution (depending on the choice of the model and the size of the domain considered). Another issue, which is related to the first one, is the variability range difference. At 40 km resolution, the observations can have a different range of values than a recording ground station, which would be more sensitive to very local events than a smoother satellite observation. Despite these issues, satellite observations still give valuable information in the spatial and the time domains.

Other studies on soil moisture assimilation have investigated the issue of the scale difference. De Lannoy et al. (2010) tested four different approaches on how to assimilate AMSR-E soil moisture products with an instrumental resolution of around 40 km, sampled on a 25 km resolution grid, into a model using a 1 km resolution grid. This work concluded that the *3D-Cm* is the





method of choice: assimilation at the observation scale (25 km) with an adjustment of the variable using a radius of influence (around 40 km). Sahoo et al. (2013) confirmed this conclusion even if it was also acknowledged that it is not the best method in terms of statistical scores, but it is the one preserving the correct spatial distribution of the model simulations (where edge effects coming from coarser observation assimilation can be avoided).

5    Assimilation is of particular interest for regions where water management is vital while in situ hydrological data are scarce. This is the case in the West African region which faces major water related risks (drought, floods, famine, diseases) threatening the population safety and slowing down the economical development. At the same time, the region is notoriously known to be lacking of in situ hydrological data which limits the possibility to properly address the water management issues.

It was already pointed out by Bitew and Gebremichael (2011) and Gosset et al. (2013) that satellite precipitation products 10 have poor to modest skills in the streamflow simulations. Gosset et al. (2013) and Casse et al. (2015) highlighted the positive bias in satellite precipitation products leading to an overestimation of the runoff and *in fine* an overestimation of the discharge when they were used in hydrological modeling. Thiemig et al. (2013) have compared different methods to correct for these biases by either correcting the products themselves or by adjusting the parameterization of the model. The objective here is to constrain the water and energy balances by assimilating surface soil moisture satellite observations despite the inaccuracy of 15 the satellite rainfall products used as forcings in the model.

Our study focuses on the assimilation of SMOS soil moisture over a West African catchment in Benin and investigates the impact on the other hydrological variables, especially the streamflow simulations. A first part of this article presents the study area and the satellite data used in the study. Then the hydrological model DHSVM is briefly described along with the assimilation method. The results of the assimilation and its impact are presented on the last section before the conclusions.

## 20  2   Study area and satellite data

### 2.1   The Ouémé catchment and the in situ measurements

The Ouémé catchment is located in Benin, West Africa, and is part of the AMMA-CATCH observatory (African Monsoon Multidisciplinary Analysis - Coupling the Tropical Atmosphere and the Hydrological Cycle, Lebel et al. (2009), *www.amma-catch.org*) whose objective is to study the hydrological impact of climate change. With a size of 12,000 km$^2$, the Ouémé 25 catchment is mainly covered by savanna, forests and cultures. The rain season spreads from April to October for an annual amount of around 1500 mm. This basin is highly instrumented in order to monitor the water cycle and the vegetation dynamic in this sub-humid region. Many stations are installed throughout the entire catchment measuring many soil and air variables, along with rain gauges, flux stations and streamflow stations.

Soil moisture is measured at three locations indicated by red crosses in Fig. 1. Every hour, Time Domain Reflectometry 30 (TDR) sensors measure the soil response to an electric pulse at various depths (from 5 cm to 1.2 m). Soil moisture values can be retrieved after correction for the soil temperature impact and by using wet and dry samples from the different ground sites. The simulations from the 1 km$^2$ model pixel containing the in situ measurements is used for the comparisons. For two of these sites, flux stations are also installed measuring the evapotranspiration every 30 minutes using eddy correlation sensors.



Water levels are measured every hour at two locations (indicated by the two red circles in Fig. 1) representing the outlets of the two sub-basins of the Ouémé catchment: Cote 238 and Beterou. For each site, a calibration has been realized to convert the water level into a streamflow value using an Acoustic Doppler Current Profilometer (ADCP). The total streamflow is supposed to be the sum of the measurements at these two points as the contribution between the real outlet of the whole basin and the

points of measurements is negligible. Comparison with simulations are also done with the sum of the simulated streamflow from these two locations.

## 2.2   Meteorological forcing data

The DHSVM model needs meteorological inputs for the following variables: relative humidity, air temperature, wind speed, pressure, shortwave and longwave radiation. The reanalysis MERRA (Modern-Era Retrospective analysis for Research and

Applications) products from NASA have been used in this study (Rienecker et al., 2011). These products are available hourly at a 1/2 and 2/3 degree resolution in latitude and longitude, and have been produced using the Goddard Earth Observing System Model, Version 5 (GEOS-5) and the Atmospheric Data Assimilation System (ADAS, version 5.2.0).

## 2.3   Precipitation data

The rainfall monitoring in the Ouémé catchment is ensured by a dense network of rain gauges (tipping bucket). For the study of

years 2010-2012, 33 evenly distributed rain gauges were operating. Their measurements have been treated in order to produce 1-hour rainfall series that have been then spatially interpolated over a regular 0.05 degree resolution grid based on Lagrangian kriging. Since the rain gauge network is dense enough, the use of interpolated rain fields to force hydrological models is relevant and can help to produce simulations of reference (Vischel and Lebel, 2007; Gascon et al., 2015). But in most cases, there are not enough rain gauges to cover the entire basin and precipitations observed by satellite can be used. Many satellite

products are available and three have been used in this study.

The PERSIANN (Precipitation Estimation from Remotely Sensed Information using Artificial Neural Networks, v. 300 and 301, Hsu et al. (1997); Sorooshian et al. (2000)) product used here is an estimation of the rainfall rate at a 0.25 degree resolution every 3 hours, based on infra-red satellite observations coupled to ground observations from gauges and radars operating at various frequencies. Some studies have already shown that this rainfall product does not perform well everywhere (Ward et al.,

2011; Thiemig et al., 2012). However, for the purpose of this paper and in order to show the benefit/impact of SMOS soil moisture assimilation, this product has been selected as a challenging case.

A second satellite product has been used for precipitation input data: the TRMM Near-Real-Time 3B42RT (v7, Huffman et al. (2010)), which combines microwave and infra-red satellite observations and is available at a 0.25 degree resolution and a 3 hour time step. This product has been widely used in various hydrological studies (Khan et al., 2011; Li et al., 2012b) and

has the advantage to combine two sources of data compared to the PERSIANN product. For the sake of simplicity, the TRMM Near-Real-Time 3B42RT product is referred as the TRMM product in the following.

CMORPH (CPC MORPHing from NOAA, Joyce et al. (2004)) precipitation data has also been tested here. This method uses rainfall estimates that have been derived from low orbit satellite microwave observations, and infrared observations from




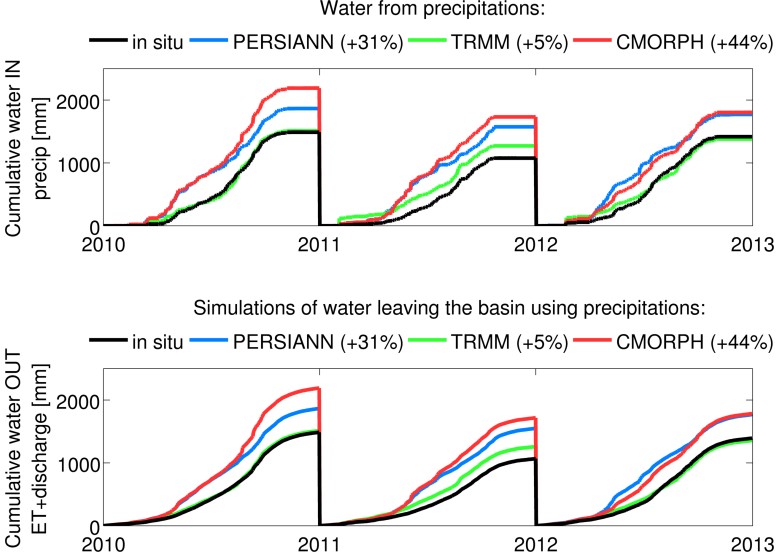

**Figure 2.** Comparison between the in situ precipitations and the three satellite products for the entire Ouémé catchment in 2010-2012 (top panel). Simulations of the amount of water leaving the basin are also compared when the different precipitation products are used (bottom panel). The additional water put into the model from the satellite products compared to the in situ is indicated between parenthesis.

geostationary satellites in order to produce a merged and unique rainfall dataset. The CMORPH product that has been selected is available at a 0.25 degree resolution and a 3 hour time step.

The three satellite products have been compared to the precipitation in situ data over the Ouémé catchment. Fig 2 shows the accumulative amount of water coming into the system (top panel) from the precipitations and leaving the basin (bottom panel) by evapotranspiration (ET) and river discharge. At the end of each year, the cumulative amounts of water that came in and left are equal since there is no variation of the water stocked into the ground. The bottom panel can be seen as a smoothing of the top panel, which can be explained by rainfall events bringing punctually a certain amount of water, whereas ET and discharge are continuous processes which make the cumulative curves look smoother.

Compared to the in situ precipitation (in black on the figures), TRMM produces 5% more volume, whereas PERSIANN 31% more and CMORPH 44% more. The largest difference between in situ and satellite rainfall happens in the second quarter of the year for the PERSIANN and CMORPH products. This time period is just before the monsoon period, which might saturate the soil earlier than it should, leading to high values of runoff and discharge at the end. For all the satellite products, the dry season is not well represented even if the rainfall amount is much lower than during the rainy season. This additional input water ingested by the hydrological model will have an impact on the output simulations. These positive biases were already identified in (Gosset et al., 2013; Casse et al., 2015).





**Table 1.** DHSVM soil and vegetation parameter values (understory and ovserstory) after calibration. The marker * indicates the parameters that have been re-calibrated compared to Gascon et al. (2015).

| Soil parameters | | Vegetation parameters | | |
|---|---|---|---|---|
| | | | Under. | Over. |
| Lateral saturated hydraulic conductivity * [m/s] | $5.10^{-2}$ | Canopy coverage [fraction] | | 0.9 |
| | | Trunk space [fraction] | | 0.4 |
| Exponential decrease rate of lateral saturated hydraulic conductivity * [-] | 2 | Aerodynamic extinction factor for wind through overstory [fraction] | | 3.5 |
| Max. infiltration rate * [m/s] | $2.10^{-4}$ | Radiation attenuation by vegetation [fraction] | | 0.5 |
| Soil surface albedo [-] | 0.1 | | | |
| Porosity * [fraction, 4 layers] | 0.5,0.5,0.5,0.5 | Vegetation height [m] | 0.5 | 6 |
| Bulk density [kg/m³, 4 layers] | 1485,1485, 1485,1485 | Fraction of shortwave radiation photosynthetically active ($R_{pc}$) | 0.108 | 0.108 |
| Field capacity * [m³/m³, 4 layers] | 0.15,0.20, 0.25,0.35 | Root zone depths [m] | 0.01,0.05,0.40,1.0 | |
| Wilting point * [m³/m³, 4 layers] | 0.02 0.04, 0.08,0.12 | SM threshold above which transpiration is not restricted [m³/m³] | 0.10 | 0.30 |
| Vertical saturated hydraulic conductivity * [m/s, 4 layers] | $10^{-7},10^{-6},$ $10^{-6},10^{-6}$ | Vapor pressure deficit threshold above which stomatal closure occurs [Pa] | 3000 | 2500 |
| Thermal conductivity [W/m.K, 4 layers] | 7.114,7.114, 7.114,7.114 | | | |
| Thermal capacity [J/m³.K, 4 layers] | $1.4.10^6,1.4.10^6,$ $1.4.10^6,1.4.10^6$ | | | |

## 2.4 SMOS soil moisture product

The Soil Moisture Ocean Salinity (SMOS) mission has been producing soil moisture products for more than five years, observing the entire globe every three days at a resolution of around 40 km. Due to multi-angular observations and the sensitivity of the L-band frequency to the soil water content, the soil moisture is retrieved with a target accuracy of 0.04 m³/m³. More

5 details can be found on the soil moisture retrieval algorithm in Kerr et al. (2012).

The SMOS Level 3 soil moisture product (2nd reprocessing, v. 2.7, 1 day product, Jacquette et al. (2010)) used in this study is provided by CNES-CATDS (*Centre Aval de Traitement des Données SMOS*) on the EASE-Grid 2.0 (Equal-Area Scalable Earth) at 25 km resolution. In Louvet et al. (2015), it was found that SMOS L3 product is the most suitable and available satellite soil moisture product compared to in situ measurements collected in West Africa from 2010 to 2012.





## 3  Model and assimilation method

### 3.1  DHSVM

For the Ouémé catchment, Seguis et al. (2011) shows a major contribution of lateral water flows in the hydrological processes, especially during the spring season. The Distributed Hydrology Soil Vegetation Model (DHSVM, developed at the University of Washington, Wigmosta et al. (1994)) has been selected for its capability of water lateral redistribution from and to the neighboring pixels.

DHSVM solves the energy and water balances at each grid cell and time step with a physically based model representing the effect of topography, soil and vegetation. The outputs are the soil moisture, the snow quantity (not used nor showed here), the streamflow, the evapotranspiration and the runoff. This model has already been used in many previous studies (Whitaker et al., 2003; Cuo et al., 2006; Cuartas et al., 2012; Du et al., 2014; Gascon et al., 2015) showing its capability to simulate various hydrological components such as the snowpack, the streamflow, the water table depths or the soil moisture. All these studies also emphasized the importance of the model parameter calibration step and the accuracy of the meteorological input data.

DHSVM has been used in this study at a resolution of 1 km with an hourly time step and four soil layers at the following depths: 1 cm, 5 cm, 40 cm and 80 cm. The first layer has been set for numerical reasons, the second is used for the assimilation, and the two deeper layers are used for validation with in situ measurements.

The DHSVM model has many physical parameters that need to be calibrated based on soil characteristics and vegetation covers. Previous studies (Whitaker et al., 2003; Cuo et al., 2006; Cuartas et al., 2012; Du et al., 2014) described precisely their DHSVM model parameter values using in situ radiation, soil moisture and streamflow measurements for calibration. It was often noticed that it was difficult to obtain good soil moisture and streamflow simulations simultaneously, and that streamflow simulations could be improved at the expense of the soil moisture simulations (Cuo et al., 2006).

In Gascon et al. (2015), DHSVM parameterization was realized using in situ streamflow measurements at the Cote 238 station for 2005. This parameterization has been used as a starting point for this study. Here, the model has different soil layers and has been calibrated using in situ measurements from 2010 (soil moisture from the three stations, streamflow at the outlet, and evapotranspiration from one station). In order to ingest the correct amount of water for the calibration process, the interpolated in situ rainfall data have been used. Table 1 represents the main soil and vegetation characteristics used in this study for the DHSVM model after calibration. These parameter values have been optimized using a semi-automatic protocol, i.e., multiple sets of values have been tested and the one giving the best performance has been chosen. Model outputs have been evaluated at different locations in the basin using soil moisture (R=0.81, RMSE=0.084 $m^3/m^3$), streamflow (R=0.94, RMSE=81.7 $m^3/s$, ME=0.87) and evapotranspiration (R=0.81, RMSE=166.7 $W/m^2$) in situ measurements.

As it has been mentioned in Bitew and Gebremichael (2011), calibrating a model using satellite precipitations will lead to a set of parameters that will compensate for the extra runoff generated by the extra volume of water brought by the satellite product compared to the in situ measurements. Adjusting the model parameters can compensate for the rainfall errors but the global water budget will be deteriorated and the other hydrological processes will be disturbed. For this reason, the model calibration has only been performed with the in situ precipitations, which leads to a correct partitioning of the precipitation




between infiltration and runoff thanks to the good quality of this dataset. It is not expected from the performances using satellite precipitation products to be as good as the ones using in situ precipitation, and only the change between before and after the assimilation will be studied.

### 3.2 Assimilation methodology

SMOS soil moisture is assimilated into the DHSVM model using an optimal interpolation method (simplification of the Kalman filter where the errors are assumed to be known). In this study, the "3D-Cm" method proposed in De Lannoy et al. (2010) and successfully used in Sahoo et al. (2013), is applied here. The "3D-Cm" scheme consists in assimilating multiple coarse scale observations (25 km), which implies an aggregation of the model from the fine scale (1 km) to the SMOS scale but avoids artificial transitions at the pixel boundaries by using multiple coarse scale observations to update the finer scale simulations.

Some of the key equations of the assimilation method are detailed in this article but more information can be found in De Lannoy et al. (2010) or in Sahoo et al. (2013).

Based on the difference between the simulations and the observations, the model background predictions are updated depending on their respective error covariances. Ensemble methods can estimate these error covariances from a Monte-Carlo ensemble generation but in this study, a simpler method has been applied and fixed values of the error covariances are used.

Before being assimilated and for an optimal analysis (Yilmaz and Crow, 2013), the SMOS soil moisture product has been rescaled to remove any systematic bias using the open-loop model simulations. The term *open-loop* refers to simulations with no assimilation. In this study, a CDF (Cumulative Density Function) matching at SMOS scale has been applied for each pixel independently for each year. Also, the ascending (6 a.m.) and the descending (6 p.m.) observations have been treated separately.

Each time step $i$ a SMOS observation is available, the forecast state vector $\hat{x}_i^-$ including the soil moisture at the four model

soil depths (1 cm, 5 cm, 40 cm and 80 cm) is mapped from the fine model scale (1 km) to the coarse SMOS scale (25 km) to calculate the prediction at the observation scale $H\hat{x}_i^-$, where $H$ is called the observation operator. As in De Lannoy et al. (2010) or Sahoo et al. (2013), a simple spatial mean is applied here. The difference between the observation and the prediction at the coarse scale, called the innovation $(y_i - H\hat{x}_i^-)$, is used to update the finer model pixels $\hat{x}_i^+$ called the analysis using a gain matrix $K$. The update equation at time step $i$ for a given fine scale pixel $k$ is as follows:

$$\hat{x}_i^{k+} = \hat{x}_i^{k-} + K_i^k \left[ y_i - H\hat{x}_i^- \right] \tag{1}$$

where the gain matrix $K$ depends on the model error covariance $B$ and the observation error covariance after rescaling $R$:

$$K = \frac{BH^T}{HBH^T + R} \tag{2}$$





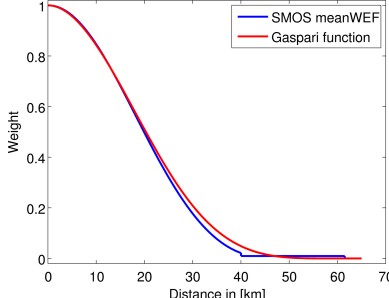

**Figure 3.** Weighing functions for the observation matrix $H$ comparing the function used for the SMOS antenna pattern and the Gaspari function used in this study.

The model error covariance matrix $B$ is calculated separately for each pixel of the model grid based on the DHSVM open-loop simulations ($B_{ij} = Cov(SM_i, SM_j)$). The average $B$ matrix is as follows:

$$B = \begin{bmatrix} 0.022 & 0.015 & 0.010 & 0.003 \\ 0.015 & 0.019 & 0.011 & 0.003 \\ 0.010 & 0.011 & 0.012 & 0.005 \\ 0.003 & 0.003 & 0.005 & 0.006 \end{bmatrix} (m^3/m^3)^2 \qquad (3)$$

The SMOS observation error covariance matrix $R$ is evaluated for each node of the SMOS grid using all the available SMOS observations. $R$ is supposed to be diagonal and represents the variance of the observations. The average variance of the SMOS observations is $0.017 \ (m^3/m^3)^2$.

Finally the observation matrix $H$ consists of 4 columns (for the four soil layers) times the number of available observations for the number of lines. Since the assimilation is performed on the second soil layer, the second column $H$ should be filled with the same equal value if all SMOS observations had the same influence on the model grid point of interest (sum of these values equal to 1). For this reason, a weighing function is used depending on the distance between the SMOS observation and the concerned model point such as shown in Fig. 3.

Here, $y_i$ contains as many SMOS observations as are within a given radius (60 km) and those observations have a larger impact if they are closer to the considered model pixel to update. As in Reichle and Koster (2003) and De Lannoy et al. (2010), a fifth-order polynomial function (Eq. (4.10) of Gaspari and Cohn (1999)) based on the distance between two points and on a compact support radius is applied to weigh the influence of SMOS observations in $H$ (Gaspari function, red line in Fig. 3). This equation is really close to the SMOS mean weighting function used to model the antenna pattern in the SMOS retrieval algorithm (Kerr et al. (2015), blue line in Fig.3).

The SMOS observations are assimilated in the second soil layer of the model (1-5 cm). The correlations between the different soil layers being contained in $K$, the other soil layers (1 cm, 40 cm and 80 cm) are also updated during the same time step but with a lower influence from the SMOS observations. The other model variables such as the evapotranspiration and the





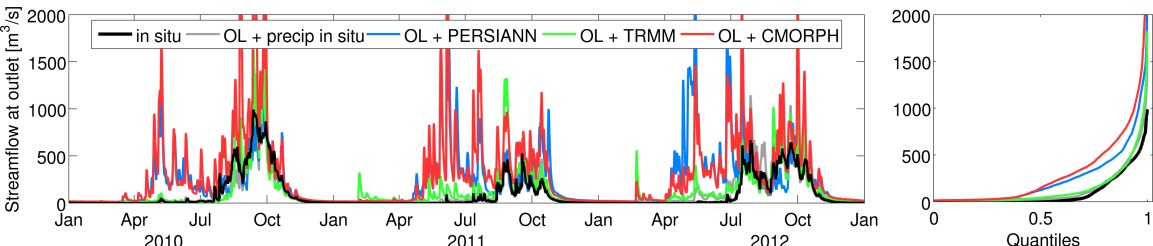

**Figure 4.** Streamflow simulations at the basin outlet for 2010-2012 using four rainfall forcing products: measurements from rain gauge stations (gray), PERSIANN (blue), TRMM (green), and CMORPH (red), compared to the in situ streamflow measurements (black). Quantiles are indicated on the right panel. Statistics are reported in Table 3.

streamflow are not updated through the assimilation step but are updated with the propagation of these modifications in the model, i.e., if water is removed from the ground, the lateral subsurface flow and the streamflow should decrease too.

### 3.3 Statistics metrics

In order to quantify the performances of the model simulations and the impact of the SMOS soil moisture assimilation, five statistics metrics have been chosen in this study: the temporal correlation R, the bias, the standard deviation of the difference between the simulations and the in situ measurements (sdd), the root mean square errors (RMSE = $\sqrt{\text{bias}^2 + \text{sdd}^2}$) and the Nash model efficiency coefficient ME as defined in Nash and Sutcliffe (1970) for streamflow simulation skill. These statistics have been computed using every common dates available.

Four precipitation products are tested in this study: in situ, PERSIANN, TRMM and CMORPH as described earlier. An other case is also investigated where no rain is given to the model in order to see the impact of the assimilation in this extreme example.

### 4 Results

#### 4.1 DHSVM open-loop simulations

Soil moisture measurements have been used for the calibration of the DHSVM parameters, especially for the vertical conductivity in order to correctly control the amount of water that can go through the different layers. Three ground stations provide soil moisture measurements and are all located in the North-Western part of the Ouémé catchment (red crosses in Fig.1). Among the three possibilities, Bira station has been chosen for the availability and the consistency of the soil moisture measurements at the three depths (5 cm, 40 cm and 80 cm). Using in situ precipitations, the correlation scores are good for the three layers (from 0.71 to 0.89, see Table 2). However the RMSE values are large (from 0.091 $m^3/m^3$ to 0.139 $m^3/m^3$) and could only be lowered at the expense of the streamflow simulations. This statement is in line with previous attempts of calibration of the DHSVM model, such as in Cuo et al. (2006), where good simulations of both soil moisture and discharge could only be obtained by





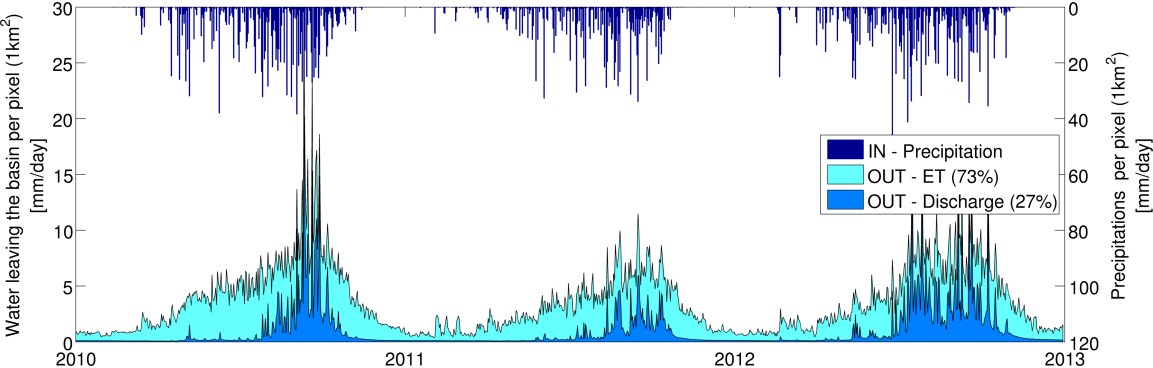

**Figure 5.** Distribution of the water for the entire basin in 2010-2012 [in mm/day] using in situ precipitations. The water leaves the basin by evapotranspiration mainly (73%) and by river discharge (27%).

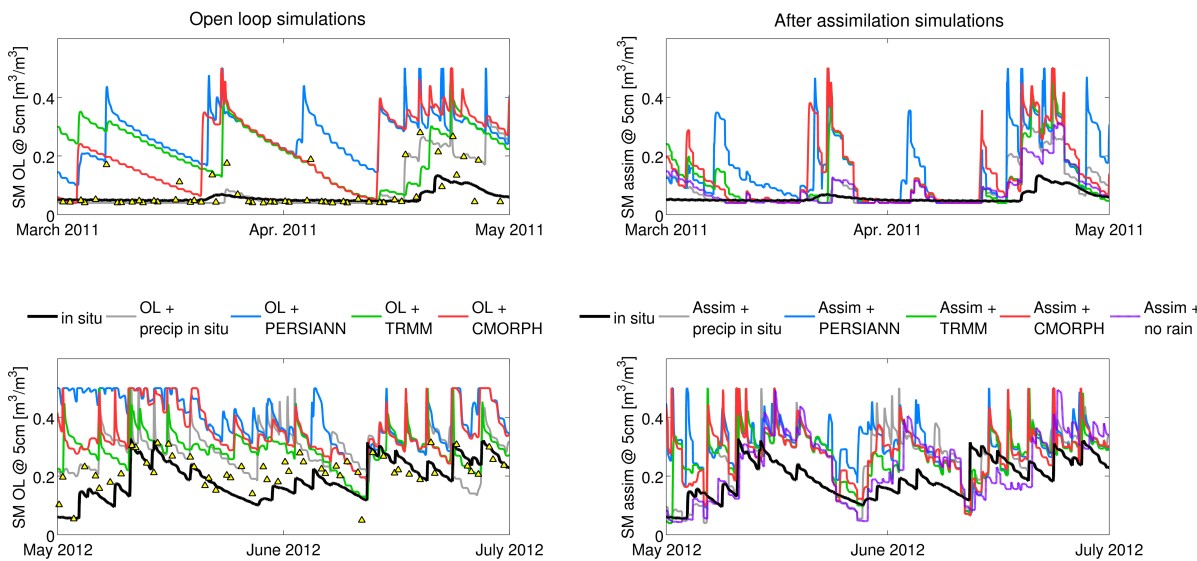

**Figure 6.** Comparison between the simulations of soil moisture at 5 cm depth at the Bira station at two different time periods: dry season in 2011 (upper panel), and the beginning of the raining season in 2012 (lower panel). The open-loop simulations are represented on the left whereas the simulated soil moisture after assimilation are on the right. The different rainfall products are indicated with various colors, except for the no-rain case using the open-loop as the soil moisture is at its lowest value. Assimilated SMOS observations are indicated by yellow triangles on the left panel.

applying unrealistic values of soil parameters. From the open-loop scores of Table 2, the results are naturally better when the in situ precipitations are used compared to the satellite rainfall products since the parameterization has been performed using in situ precipitations.





The in situ streamflow measurements have also been used for the calibration of the DHSVM parameters. Fig. 4 shows the good fit between the simulated streamflow for 2010-2012 using the in situ rainfall data and the streamflow measured at the outlet of the basin with a correlation of 0.92 and a Nash coefficient of 0.73. Fig. 4 also shows the difference between the simulations when in situ, PERSIANN, TRMM, or CMORPH precipitation products are used (Table 3 for the open-loop

statistics). PERSIANN and CMORPH products exhibit strong precipitation events during the Spring season (April to June), so streamflow increases in this period of the year whereas in situ measurements (and TRMM to a lesser extend) do not show any discharge. For this reason, PERSIANN and CMORPH statistics are quite poor with negative Nash model efficiency coefficients. TRMM performs better, giving reasonable amount of water to the model as inputs, leading to also reasonable streamflow simulations with a correlation of 0.86 and a Nash coefficient of 0.47 for 2010-2012. Since the simulations of the streamflow

can change very quickly due to the intense rainfall events, the right panel of Fig. 4 shows the quantiles for the five cases. The discrepancies stated before can clearly be seen here with PERSIANN and CMORPH curves above the others.

Another important aspect in hydrological modeling is the repartition of the precipitations into the possible different model compartments. Fig. 5 shows the time series of the precipitations (incoming water), the evapotranspiration and the basin discharge outflows. At the end of the year, the sum of the outputs is equal to the input as the water stock in the soil is negligible

(less than 1% for the three years of the study). With the calibrated DHSVM parameterization, it is found that 73% of the water leaves the catchment through evapotranspiration and 27% by river discharge (mainly during the rainy season since there is no streamflow during winter).

From this point, these simulations are used as benchmarks for the next step of data assimilation. Statistics after SMOS assimilation will be compared to the open-loop simulations in order to see the impact of the assimilation.

## 4.2   Impacts of SMOS soil moisture assimilation

### 4.2.1   Soil moisture simulations at the Bira station

The first variables to be impacted by the assimilation of SMOS products are the ones directly contained in the state vector of the assimilation scheme, i.e. the soil moisture of the the four defined soil layers at 1 cm, 5 cm, 40 cm and 80 cm. Soil moisture simulations are shown in Fig. 6 at 5 cm depth for two time periods: the upper panel represents the time series of March-April

2011 (beginning of the rain season), and the lower panel May-June 2012 (wet season). The left side shows the open-loop simulations whereas the after-assimilation results are on the right side. For visual clarity, the three years of simulations are not shown here but these two time periods are representative of the effect of assimilation on the soil moisture variable.

As mentioned before, the satellite rainfall products bring too much water during the winter and spring seasons. The first time period is a good example of a soil moisture increase after a rainfall detected by the satellite product (at the beginning of March

for example) which is not present in the in situ rain product. The simulated soil moisture is thus impacted by this *fake* rain event. By assimilating SMOS soil moisture product at the surface, the impact of this wrong rainfall event is smoothed but has not completely disappeared. The wet season example also shows the same process. Peaks in the soil moisture time series due





**Table 2.** Statistics of the simulated soil moisture at 3 depths (5, 40 and 80 cm) from the open-loop (O-L) and the assimilation (Assim.), compared to in situ measurements at the Bira station for 2010-2012. Bias, standard deviation of the difference (sdd) and root mean square error (rmse) are in $m^3/m^3$, the correlation (R) is dimensionless. Bold font is used when the bias is improved by the assimilation.

| SM (5cm) | In situ precip. | | PERSIANN precip. | | TRMM precip. | | CMORPH precip. | | *No rain* |
|---|---|---|---|---|---|---|---|---|---|
| | O-L | Assim. | O-L | Assim. | O-L | Assim. | O-L | Assim. | *Assim.* |
| R | 0.82 | 0.81 | 0.60 | 0.73 | 0.72 | 0.81 | 0.76 | 0.78 | *0.81* |
| bias | 0.049 | 0.046 | 0.091 | **0.062** | 0.051 | 0.051 | 0.089 | **0.056** | *0.030* |
| sdd | 0.085 | 0.077 | 0.119 | 0.091 | 0.098 | 0.082 | 0.102 | 0.086 | *0.074* |
| rmse | 0.098 | 0.090 | 0.150 | 0.110 | 0.110 | 0.096 | 0.136 | 0.103 | *0.080* |
| (40 cm) | | | | | | | | | |
| R | 0.89 | 0.63 | 0.62 | 0.65 | 0.75 | 0.72 | 0.76 | 0.67 | *0.80* |
| bias | 0.062 | 0.062 | 0.119 | **0.052** | 0.086 | **0.071** | 0.128 | **0.064** | *0.144* |
| sdd | 0.067 | 0.087 | 0.085 | 0.099 | 0.068 | 0.094 | 0.072 | 0.101 | *0.059* |
| rmse | 0.091 | 0.107 | 0.146 | 0.112 | 0.110 | 0.117 | 0.147 | 0.120 | *0.156* |
| (80 cm) | | | | | | | | | |
| R | 0.71 | 0.32 | 0.64 | 0.49 | 0.52 | 0.42 | 0.69 | 0.50 | *0.55* |
| bias | 0.113 | 0.127 | 0.194 | **0.102** | 0.154 | **0.136** | 0.200 | **0.131** | *0.214* |
| sdd | 0.081 | 0.125 | 0.064 | 0.126 | 0.083 | 0.115 | 0.056 | 0.119 | *0.064* |
| rmse | 0.139 | 0.178 | 0.204 | 0.162 | 0.175 | 0.178 | 0.208 | 0.177 | *0.223* |

to an inaccurate amount of precipitation cannot be corrected by the assimilation but the decreasing phase can be fastened as a post-event correction.

Table 2 compiles the statistic scores of all the precipitation cases (open-loop and after assimilation) for the three years and the three layers at the Bira station. As it can be seen in Fig. 6, the continuity in the soil moisture time series cannot always be preserved by the assimilation method applied here, which results in abrupt changes before and after the time step when the assimilation is performed. This discontinuity has a negative impact on the statistical scores such as the correlation, the standard deviation, and the root mean square error. The bias is the only statistical metrics that can be used to truly assess the impact of the assimilation on the soil moisture variable. The other statistics are shown for indication.

Regarding the satellite precipitation products, the bias is always reduced after the assimilation. At 5 cm depth, it is improved by 0% (TRMM) up to 37% (CMORPH), at 40 cm depth by 17% (TRMM) up to 56% (PERSIANN), and at 80 cm, by 12% (TRMM) up to 47% (PERSIANN). This shows that the assimilation and the model are able to propagate the information from the 5 cm layer to the deeper layers of the soil. The largest improvements are naturally obtained when the PERSIANN and the CMORPH products are used as precipitation forcing. Assimilation can correct for the additional amount of water brought by these two satellite products at the soil level. Moreover, the open-loop simulations show unrealistic soil saturation at the 5 cm





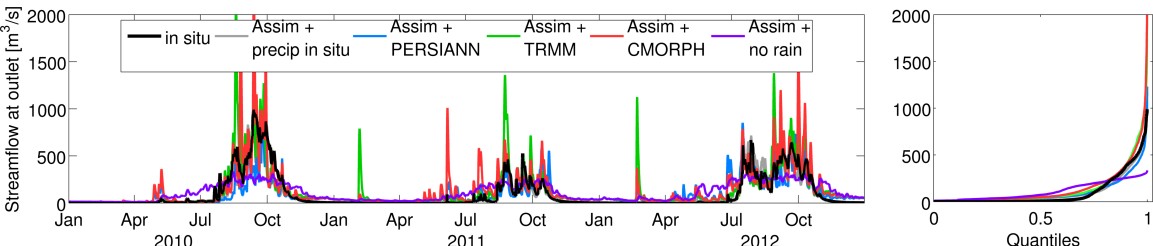

**Figure 7.** Streamflow simulations at the basin outlet for 2010-2012 after SMOS assimilation using the four rainfall forcing products: measurements from rain gauge stations (gray), PERSIANN (blue), TRMM (green), and CMORPH (red), compared to the in situ streamflow measurements (black). An extra case is also added where no precipitation data is used and the only source of water comes from the assimilation (purple). Quantiles are indicated on the right panel. Statistics are reported in Table 3.

layer during the rain season (soil moisture value is equal to porosity, see Fig. 6), which is also the case at deeper layers later in the season (not shown here). This saturation issue is improved after assimilation, but can still happens.

Assimilation does not correct directly the precipitations: neither for the amount of water nor for the time of the event itself. So the volume of water given to the model remains the same and the peaks in the soil moisture simulations cannot be corrected
until a SMOS observation becomes available, and only the decreasing phase can then be modified.

The extreme case where no rainfall information is available only relies on the soil moisture assimilation. The statistics show a very good agreement with the in situ measurements at 5 cm, where the assimilation is performed. The correlation is high (0.81) and the bias value is the lowest with 0.030 $m^3/m^3$. The statistics scores weaken for the deeper layers but are not the poorest regarding the temporal statistics (correlation R and standard deviation of residuals sdd). This means that surface soil
moisture assimilation is also able to bring reasonable information regarding the amount of water in the soil.

### 4.2.2 Streamflow simulations at the basin outlet

Even if the data assimilation scheme only adjusts the water contained in the different soil layers, the soil water content interacts with the surface and sub-surface processes and thus influences the streamflow at the end.

Simulations of the streamflow (open-loop and after assimilation of the SMOS soil moisture products) are compared to the
in situ measurements at the outlet of the catchment. The streamflow simulations after assimilation are shown in Fig. 7 and the statistics are indicated in Table 3.

Compared to the open-loop simulations in Fig. 4, improvements can clearly be identified: the spikes are smoother, the dry season is more respected, and the time evolution is less chaotic. PERSIANN and CMORPH products identify precipitation events from March to July that have not been observed by the in situ rain stations (see Fig. 2) and this additional water is found
back in the streamflow output simulations. In general, PERSIANN and CMORPH bring more water into the model, which tends to produce these spikes in the streamflow simulations. After assimilation of the SMOS soil moisture products, the peaks are smoother and the general time variation is much more in line with the in situ observations. The assimilation of a surface variable has here a real impact on the hydrology at a catchment scale.





**Table 3.** Statistics of the simulated streamflow at the outlet compared to the in situ measurements for the open-loop case (O-L) and for the assimilation case (Assim.). Bias, standard deviation of the difference (sdd) and root mean square error (rmse) are in $m^3/s$, the correlation (R) and the model efficiency (ME) coefficients are dimensionless with 1 representing a perfect match.

| 2010-2012 | In situ precip. | | PERSIANN precip. | | TRMM precip. | | CMORPH precip. | | *No rain* |
| --- | --- | --- | --- | --- | --- | --- | --- | --- | --- |
| | O-L | Assim. | O-L | Assim. | O-L | Assim. | O-L | Assim. | *Assim.* |
| R | 0.92 | 0.90 | 0.40 | 0.78 | 0.86 | 0.81 | 0.64 | 0.81 | *0.75* |
| bias | 31.8 | 7.4 | 144.1 | 4.5 | 44.5 | 40.9 | 212.3 | 47.8 | *18.0* |
| sdd | 85.2 | 75.9 | 286.9 | 111.2 | 120.3 | 131.4 | 355.5 | 134.2 | *123.1* |
| rmse | 90.9 | 76.2 | 321.1 | 111.3 | 128.2 | 137.6 | 414.1 | 142.5 | *124.4* |
| ME | 0.73 | 0.81 | -2.33 | 0.60 | 0.47 | 0.39 | -4.53 | 0.35 | *0.50* |

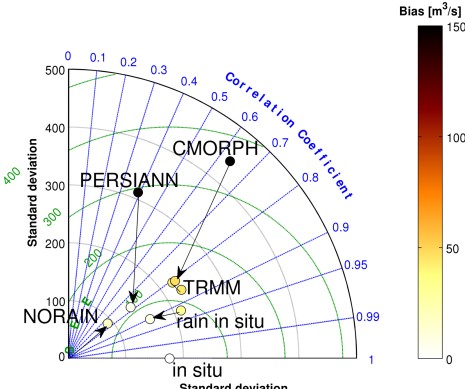

**Figure 8.** Taylor diagram of the streamflow simulations at the outlet of the Ouémé catchment. The arrows indicate the change in the statistics from the open-loop to the assimilation simulations using different precipitation products : in situ, PERSIANN, CMORPH, TRMM, and no rain product.

The extra input water can then be corrected by soil moisture assimilation, which has an impact on the soil moisture values of all the layers in a first step, but also on the runoff, the subsurface lateral flow, followed by the evapotranspiration and the amount of water intercepted by the channels. SMOS assimilation using in situ precipitations improves slightly the performances, with a Nash model efficiency (ME) of 0.73 and 0.81 for the open-loop and the assimilation respectively. There is no evidence of improvement from the assimilation using the TRMM precipitation product. Even if the dry season is mostly respected, the spikes tend to be amplified during the rainy season with an abnormal peak in February 2011. The improvements are however more visible when the PERSIANN or the CMORPH precipitation products are used. Temporal correlation after assimilation is around 0.80 for both products, RMSE is divided by 3, and the ME is now positive (0.60 for PERSIANN and 0.35for CMORPH).

Another representation of these statistics is the Taylor diagram in Fig. 8. It shows in a more graphical way the improvement brought by the assimilation of SMOS soil moisture products. The in situ circle on the bottom axis represents the point to reach



**Table 4.** Distribution of the total amount of water for the entire basin in 2010-2012 [in m]. The output is the sum of the water leaving the basin by river discharge (dis.) and by evapotranspiration (ET). The upper half concerns the open-loop (OL) simulations with statistics compared to simulations using in situ precipitations. The bottom half concerns the assimilation results with the additional case using no rainfall product. The variation of the amount of water stocked in the soil at the end of the three years is not significant (less than 1%) and is not indicated in this table.

| Precip. Prod. | $\sum$ dis. [m] | + | $\sum$ ET [m] | = | output [m] |
|---|---|---|---|---|---|
| *open-loop* | | | | | |
| in situ | 1.08 | + | 2.87 | = | 3.95 |
| PERSIANN | 2.08 | + | 3.13 | = | 5.18 |
| | *(+89%)* | | *(+9%)* | | *(+31%)* |
| TRMM | 1.18 | + | 2.95 | = | 4.13 |
| | *(+9%)* | | *(+3%)* | | *(+5%)* |
| CMORPH | 2.63 | + | 3.06 | = | 5.69 |
| | *(+144%)* | | *(+7%)* | | *(+44%)* |
| *after assimilation* | | | | | |
| in situ | 0.87 | + | 3.06 | = | 3.93 |
| | *(-19%)* | | *(+6%)* | | *(-1%)* |
| PERSIANN | 0.84 | + | 3.14 | = | 3.98 |
| | *(-22%)* | | *(+9%)* | | *(+1%)* |
| TRMM | 1.15 | + | 3.12 | = | 4.27 |
| | *(+6%)* | | *(+9%)* | | *(+8%)* |
| CMORPH | 1.20 | + | 3.12 | = | 4.32 |
| | *(+11%)* | | *(+9%)* | | *(+9%)* |
| no rain | 0.96 | + | 3.17 | = | 4.13 |
| | *(-11%)* | | *(+10%)* | | *(+5%)* |
| *(statistics compared to OL simulations using in situ precipitations)* | | | | | |

by the simulations, which would mean that there is a temporal correlation of 1 (blue radial axis on the right), that the standard deviation is the same as the in situ (temporal variability is then the same, gray circular axis), that the standard deviation of the difference between the simulations and the in situ is null (green semi-circular axis), and that the bias would also be null (point circle filled with colors indicated by the color bar on the right). In other words, the closest to the in situ point, the better.

5    The arrows on the diagram show the impact of the assimilation on the statistics of the open-loop. When using the in situ rain product, the after-assimilation point is not much closer but the color indicating the bias is lighter showing an improvement in the bias. As mentioned before, simulations using PERSIANN and CMORPH products are greatly improved by the assimilation attested by these long arrows ending much closer to the in situ point. TRMM points are really close to each other showing no





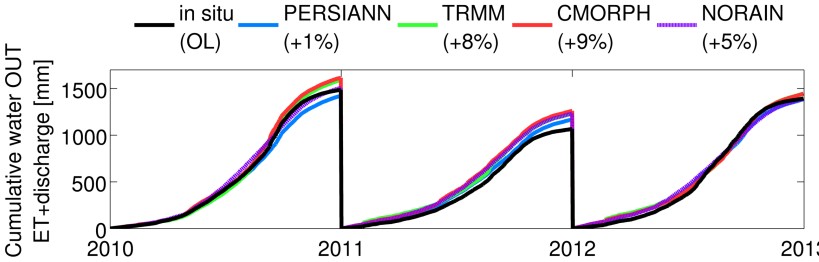

**Figure 9.** Output amount of water for different SMOS assimilation cases : using PERSIANN, TRMM, CMORPH and no rain products.

evidence of improvement from the assimilation but it should also be mentioned that the simulations before the assimilation already give statistical results as good as PERSIANN and CMORPH after the assimilation process.

The last case uses the *no rain* scenario. The only incoming water comes from the increments of water brought by the assimilation method and the updates of soil moisture at the different layers as there is no precipitation in this case. Unlike the assimilation, the open-loop simulations of the streamflow are null and do not appear on the statistics tables or on the Taylor diagram. Fig. 7 shows the time series of the simulated streamflow after assimilation, which does not have a good time variability compared to the satellite rainfall products. But the corresponding statistics in Table 3 are not as poor as expected, with a time correlation of 0.75, a bias of 18 m$^3$/s for a total RMSE of 124 m$^3$/s, and a ME of 0.50. These scores are sometimes even higher than other using a forcing precipitation product. On the Taylor diagram (Fig. 8), the *no rain* point is not far from the in situ point. This test case clearly shows the ability to bring water into the model by assimilation in the scenario that no rainfall product is available.

### 4.2.3 Other water cycle variables

Another important aspect of using an hydrological model is the water distribution in the basin. The input water volume comes from the precipitations, and leaves the basin by river discharge or by evapotranspiration. Water stays in the soil but the variation of its volume at the end of the year is extremely small (less than 1%). Table 4 summarizes the water budget of 2010-2012 (average cumulative amount of water for each pixel). The upper part of this table shows the distribution of the water for the open-loop simulations using the four precipitation products: in situ, PERSIANN, TRMM and CMORPH. The first line (representing the water budget components using the in situ precipitations) is used as the reference for the statistics between parenthesis.

Despite the large difference between the different precipitation products, the output evapotranspiration component is not highly impacted (from +3% to +9%), probably due to a sufficient volume of water already contained in the soil. On the contrary, the streamflow simulations vary much more, from +9% to +144%. Most of the additional water (brought by the satellite precipitation products compared to in situ) goes to the river flow, which is why the statistics scores are lower in Table 3.





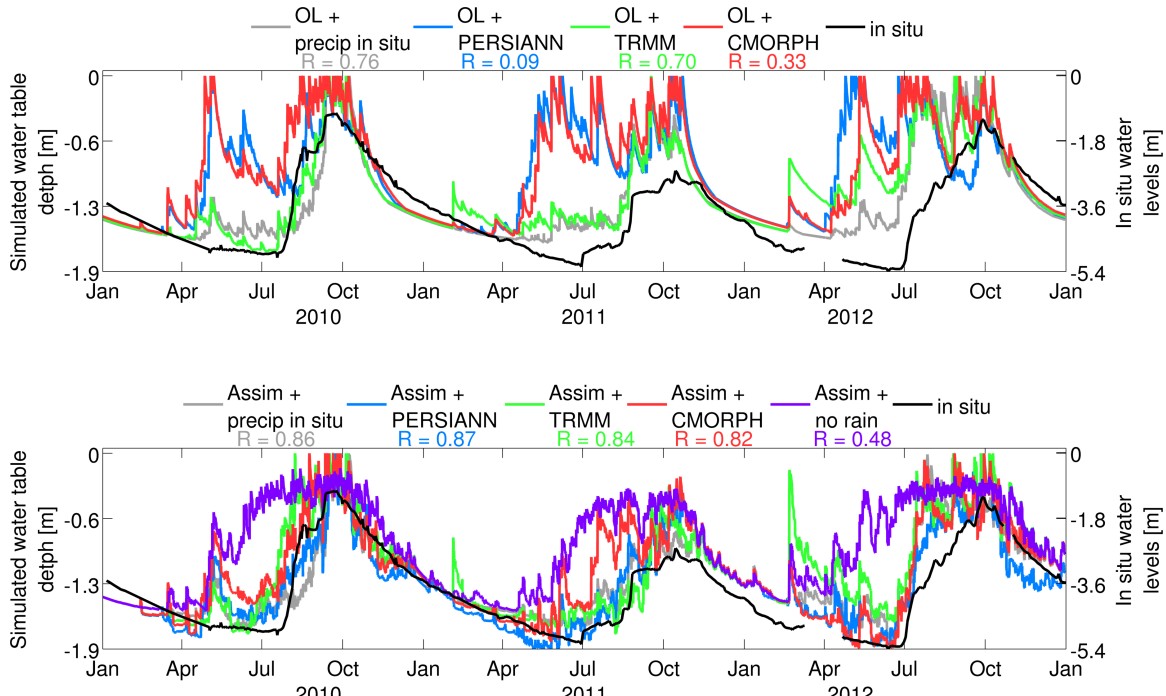

**Figure 10.** Water table depth simulation for the open-loop and the assimilation cases using the different precipitation products compared to the in situ water level from a deep well. Temporal correlation scores indicate the benefit of the assimilation on the simulated variables at deeper soil layers.

The second part of the table concerns the after-assimilation results. The incoming volume of water given to the hydrological model is the same as for the open-loop cases. But after each assimilation step, the soil moisture values are updated, which has an impact on the streamflow and on the evapotranspiration. In the following, we compare the output volume of water after assimilation with the in situ precipitations. As soil moisture assimilation is a correction of what is ingested by the model (i.e. a correction of the precipitations), comparing these two amounts of water is a possible way to measure the impact of the assimilation on the whole water cycle of the basin.

Compared to the open-loop simulations, the evapotranspiration is not highly impacted by the assimilation process. The main changes happen in the volume of water contributing to the river discharge. They are all converging to the values of the open-loop simulations using in situ precipitations. The in situ precipitation case helps to identify any bias in the system. Here, the after-assimilation sum is 3.93 m whereas the real incoming volume of water is 3.95 m, which represents a slight decrease of 1%. The three satellite precipitation cases produce output volumes of water of 3.98 m , 4.27 m, and 4.32 m respectively for PERSIANN, TRMM, and CMORPH. Except for TRMM, assimilation improves the total volume of water leaving the basin. TRMM stays however not far away from the in situ reference. At the end, using PERSIANN with SMOS assimilation puts





almost the same amount of water in the model as it should compared to the in situ precipitations, +8% using TRMM, and +9% for CMORPH.

The option using no rainfall product is also very close to the in situ volume value with only an increase of 5%. In this case, the evapotranspiration is higher than the others whereas the streamflow is underestimated. However, as seen in the previous

section, the time variability of the streamflow is not well simulated even if the total volume is coherent. This shows that the correct amount of water is given by the assimilation but the distribution (in time and space) is not good enough to represent the time dynamics of the water cycle in this case. A soil moisture product with a finer resolution in time and/or space might improve the results.

Fig. 9 shows the accumulation of the water leaving the basin per year after the assimilation compared to the in situ input water

(in black). Compared to Fig.2, the cumulative volumes of water using the different precipitation products are not overestimated anymore and follow an equivalent time evolution throughout the year. The four assimilation cases tend to simulate however the same volume of water, which is a good sign of consistency of the method.

Another important hydrological variable simulated by DHSVM is the water table depth. It represents the sub-surface limit between unsaturated and saturated soil. Groundwater is an important resource, especially in West Africa where most of the

drinking water comes from the ground and where some of the agricultural practices use irrigation. Moreover, the interannual variability of precipitation can be important (1560 mm in 2010 followed by only 1100 mm in 2011 and 1450 mm in 2012 from the in situ rain gauge measurements), which has a strong impact on groundwater recharge. The water table depth varies between the soil depth and the ground surface (in the latest case, a flooding can happen). In this parameterization of DHSVM, the soil depth for the Ouémé catchment has been fixed to 1.90 m after several spinup runs during the model calibration phase.

The water table depth does not depend on the soil depth but it depends on the model capability to evacuate this saturated water through the defined hydrological network, which involves the river channel heights (set also to 1.90 m). With this configuration, all the water going in the direction of a stream will be captured by the river channel and no water can be stocked below.

Simulations of the water table depth are available for the study period 2010-2012 and can be compared to the measurements of the water level from a deep well located at Nalohou Seguis et al. (2011) (Fig. 10), not far from the Bira station (less than

20 km). This station has been selected for the availability of its measurements along the three years of study, which have been compared to simulations from the corresponding model pixel. Water table depths and water levels from wells are not quite comparable but they should follow the same time evolution (certainly because of the difference in porosity values set in the model and what is observed in reality). In order to compare both quantities, they are represented on the same graph but not at the same scale. The left y-axis represents the depth of the water as simulated by the model with a maximum of 1.90 m

(corresponding to the soil depth), whereas the right y-axis represents the in situ water level as measured in the deep well with a maximum of 5.40 m. Correlation scores are not impacted by scaling and they are indicated directly on the figure.

Fig. 10 shows the simulations compared to in situ measurements in the open-loop case (upper panel) and after SMOS assimilation (bottom panel). Here we clearly see the benefit in the deeper layers of the soil of the SMOS assimilation. The peaks in the period from April to June are strongly reduced and the temporal behavior is in line with the in situ. The correlation

scores are also a good indicator of the improvement brought by the assimilation and it is improved for all the precipitation





cases. The biggest improvement is realized when the PERSIANN precipitations are used (the poorest product among the three tested here): R=0.09 in the open-loop case, and R=0.87 after the assimilation of SMOS soil moisture product. The assimilation in the no rain case tends however to overestimate the water table depth since it increases too early in the season.

## 5  Conclusions

The rainfall-runoff model DHSVM has been calibrated for the Ouémé catchment using in situ measurements of 2010. Using rainfall data from the several rain gauges located throughout the entire basin, the simulations show a good agreement when compared to in situ measurements of soil moisture at three depths (5 cm, 40 cm, and 80 cm), of the water table depth, and of the streamflow at the outlet. Because very few basins are as well equipped in rain gauges, satellite precipitation products are used most of the times in Africa. In this study, three rainfall products are tested: PERSIANN, TRMM, and CMORPH.

Unfortunately, these products are not as accurate as in situ rain gauges and bring too much water into the model (+31%, +5%, +44% respectively), which impacts the soil moisture simulations (soil layers are quickly saturated) and the streamflow (high peaks during spring season).

This study assesses the impact of the SMOS soil moisture assimilation into the DHSVM hydrological model over the Ouémé catchment in order to correct for the wrong water input brought by the satellite precipitation products. Through an

optimal interpolation, the soil moisture at all depths are adjusted depending on the difference between the simulations and the observations from SMOS. The assimilation is performed at 5 cm depth, and the other soil layers are also impacted but to a lesser extent. The hydrological model propagates these changes from the soil layers to the other hydrological components, such as the water intercepted by the river channels, or the water table depth.

After the assimilation of SMOS soil moisture, the amount of water contained in the different soil layers is greatly improved.

Compared to the in situ measurements, the bias is decreased at all layers.

The assimilation process is able to correct for the soil moisture at the time of the observations. In other terms, when there is an inaccurate precipitation event, the soil moisture value naturally increases and the assimilation only comes afterward. When the soil moisture values are corrected, the assimilation, as it has been implemented here, does not preserve the continuity in the time series which can lead to degraded statistics scores measuring the temporal correlation or the standard deviation of the

difference of the error. Nevertheless, the RMSE using the three satellite precipitation products after assimilation are at least equivalent to the open-loop cases (TRMM) or lower (PERSIANN and CMORPH), which shows the positive impact of the assimilation on the soil moisture simulations at all depths.

The assimilation has also an indirect but large positive impact on the streamflow simulations. By adjusting the water volume contained in the soil, the water flow at the surface and sub-surface is modified, which impacts at the end the amount of water

discharge. The streamflow simulations are greatly improved. The water volume brought by the satellite precipitation products at the beginning of the rain season is decreased and the streamflow peaks are smoothed at the same time. Except for the TRMM product, the Nash model efficiency coefficients are improved (from negative to positive values for the PERSIANN and CMORPH products). The same behavior and improvement can be seen for the water table depth simulations.





An additional test case is performed using no rainfall product. The only water input comes from the assimilation process itself when the soil moisture values are adjusted. This would represent the extreme case where no rainfall information is available. The statistical results of the soil moisture simulations are not as low as one could expect (even better at 5 cm depth than using any other precipitation product) even if some precipitation events must have been missed because of the nature of the assimilation process itself (SMOS observations are available twice a day maximum). The streamflow statistics scores are in the average of the others but the temporal dynamics is not very well respected: an overestimation during the dry season and an underestimation during the wet season. At the end, the total volume of water that is brought into the system is close to the reality (4.13 m per pixel for 2010-2012, whereas in situ precipitations indicate a total amount of 3.95 m).

Several caveats should however be noticed here. The objective of this paper was to study the repartition of the precipitation between ET and discharge, and the model has been calibrated in this framework. One could argue the lack of its representativeness of what really happens at the local scale. For example, Seguis et al. (2011) studied a small transect of the Ouémé catchment (less than 2 km) and found that groundwater and rivers were disconnected, with soil depth that could go down to 20 meters. In this study, DHSVM has been chosen, calibrated and validated as a global modeling of the basin, and therefore should not be used to monitor local aspect of the water budget as it has been implemented in this study. This difference of representativeness can be seen in the last part of this study where the water table depth simulations and the groundwater measurements are compared. In this case, the amplitudes are different certainly because of the difference in soil porosity that are very difference between the parameterization (which is acceptable at the basin scale) and what is observed in situ.

The calibration of the DHSVM model has been realized using in situ precipitation observations as inputs and soil moisture and streamflow measurements of 2010. As any other calibration process, the equifinality is a real issue as several sets of parameters can lead to equivalently good simulations. Here the set of parameters has been optimized and selected as the most realistic and physically correct. The assimilation method that has been implemented here is a very simple version of the Kalman filter where the error covariances are being calculated using open-loop simulations and time series of observations. The error covariances of the simulations and the observations were around the same value (0.019 and 0.017 $(m^3/m^3)^2$ respectively), i.e., the simulations and the observations have had equivalent weights in the adjustment process. In order to avoid these hypothesis, using an ensemble Kalman filter is advised, but also requires to set variances for the ensemble generation, which could also lead to inconclusive results if not set correctly. The further step is the use of a more sophisticated assimilation method, such as the ensemble Kalman filter, in order to get a closer evaluation of the error covariances of the model and of the observations.

*Acknowledgements.* The authors would like to thank first Dr. Luc Seguis (IRD/HydroSciences, Montpellier, France) and the *Direction Générale de l'Eau du Bénin* for the streamflow measurements at the two sub-catchments (Beterou and Cote 238), and the Centre National d'Etudes Spatiales (CNES) TOSCA program for funding this project. The authors would also like to acknowledge the Global Modeling and Assimilation Office (GMAO) and the GES DISC for the dissemination of MERRA data, the AMMA-CATCH team for providing the in situ measurements, and the ALMIP-2 team for the first DHSVM calibration set.



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
