# Peer review of "Assimilation of SMOS soil moisture into a distributed hydrological model and impacts on the water cycle variables over the Ouémé catchment in Benin"

_Hydrology and Earth System Sciences, 2015_

## Referee Comment (RC1) · Anonymous Referee #1 · 29 Feb 2016

This study aims at assimilating SMOS soil moisture observations to correct for errors in precipiation and reduce uncertainties in simulated discharge for a catchment in Benin. The study uses 4 precipitation datasets to capture the uncertainties in the simulated soil moisture. This is a nice detailed study that goes beyond the standard work and should be considered for publications. However, I do feel that the authors can make some improvements and therefore recommend the revisions below. One more general remark is that the goals seems to be to improve streamflow simulations, while the paper reads more like a paper that does precipitation correction using satellite soil moisture and thereby obtain improved simulations.

Major remarks

[Figure]

Page 3, Line 8-13 Some recent studies have actually used assimilated SMOS soil moisture at a scale comparable to the catchment in the work (Lievens et al 2015 and Wanders et al 2014). These studies show that the assimilation of SMOS has a positive impact on the streamflow estimations in some scenarios. It is stated in the manuscript that it has no impact, which is contradicted by the above mentioned studies. I think a comparison with these studies would be valuable for the reader.

Some restructuring of the introduction would help to more clearly state the research gap that this paper would like to fill. I my opinion, the most novel thing done in this study is the use of multiple precipitation forcing product and the impact of SMOS DA on the hydrological simulations with these products. Now it states (Page 4, line 16-17) that SMOS assimilation impact on streamflow is the main goal, while in Page, Lines 13 it was stated that assimilation of SMOS has no impact on the streamflow performance

Page 5 Line 25-26 Using rainfall satellite product doesn't make it challenger. If one would use perfect rainfall data the potential of SMOS for streamflow improvements would be almost zero, while if the rainfall is very imperfect the potential impact is significant since the initial guess is far of and the potential improvement is large. Please remove or correct this incorrect statement.

Page 5 Line 31, Why is the 3B42RT product used instead of the reanalysis product of TRMM, which is gauge corrected and therefore has a higher quality compared to reality.

Table 2, how is it possible that the quality of the SM simulations after assimilation show a decreased performance compared to before assimilation. Does this mean that SMOS and the observations are not well aligned or is the DA procedure sub-optimal? Does result is at least counter intuitive to what one would expected after DA of additional observations.

I think some maps of the spatial improvement of the simulations would help the reader to get a better feeling with regard to where the largest potential is for further improvement. Is it the upstream areas or are better results obtained in other regions.

Minor remarks

With respect to the precipitation corrections that are in a way done I think it would be useful to mention here some studies that focus on this aspect (e.g. Crow & Bolten, 2007; Crow et al. 2011; Pellarin et al 2013; Wanders et al 2015)

Figure 1, the quality of the image low in my version of the manuscript

Page2, Line 14 in space should be spatial

Page 2, Line 21, 0.04 is only the mission requirement of SMOS and not its actual accuracy or provide a reference to confirm this.

Table 3 This table tells me with far from perfect precipitation one can gain a lot from the assimilation of satellite derived SM data, while if the forcing is almost perfect the assimilation of SM is a difficult and potentially low gain approach. Maybe some of this should be mentioned in the dicussion

Table 4 The % are not well explained in the caption of the table, please adjust. TRMM after assimilation, should that not be -6%?

References

Crow, W. T., & Bolten, J. D., 2007. Estimating precipitation errors using spaceborne surface soil moisture retrievals. Geophysical Research Letters, 34.

Crow, W. T., van den Berg, M. J., Huffman, G. J., & Pellarin, T., 2011. Correcting rainfall using satellite-based surface soil moisture retrievals: The soil moisture analysis rainfall tool (smart). Water Resources Research, 47(8), W08521.

Hans Lievens, S Kumar Tomer, Ahmad Al Bitar, GJM De Lannoy, Matthias Drusch, Gift Dumedah, H-J Hendricks Franssen, YH Kerr, Brecht Martens, Ming Pan, JK Roundy, Harry Vereecken, JP Walker, EF Wood, NEC Verhoest, VRN Pauwels, 2015, SMOS

soil moisture assimilation for improved hydrologic simulation in the Murray Darling Basin, Australia, Remote Sensing of Environment

Pellarin, T., Louvet, S., Gruhier, C., Quantin, G., & Legout, C., 2013. A simple and effective method for correcting soil moisture and precipitation estimates using amsr-e measurements. Remote Sensing of Environment, 136, 28–36.

Wanders, N., D. Karssenberg, A. de Roo, S. M. de Jong, and M. F. P. Bierkens 2014, The suitability of remotely sensed soil moisture for improving operational flood forecasting, Hydrology and Earth System Science

Wanders, N., Pan, M., Wood, E.F. 2015, Correction of real-time satellite precipitation with multi-sensor satellite observations of land surface variables, Remote Sensing of Environment, 160, 206–221

---

## Short Comment (SC1) · 3 Mar 2016

This is an interesting work, illustrating how satellite observation of rainfall and soil moisture can be complementary. However the objective of the work should be better explained and the choice of using only non-adjusted (or RT) satellite rainfall products should be better justified. If the objective of this work is to propose an alternative bias-correction method for RT satellite rainfall than the operationnal advantage of the proposed method should be developped. If the objective is to show how combining information on both rainfall and soil moisture can help a better understanding/modelling of hydrological processes, than the point would be strenghened by adding post-adjusted products (such as 3B42V7) in the study.

-is the SMOS based bias correction potentially available with better delay than what is currently done based on gauges (for instance to correct 3B42RT into 3B42v7 ) ? what are the current/future perspective on soil moisture monitoring and would the expected sampling allow for using soil moisture based bias correction to be used operationnally ? -One of the tested product (PERSIANN) has been shown by many previous authors (cited in the present paper) to have a large and steady positive bias over the region. Simple method (like pdf matching based on gauges series used by Thiemig et al, among others) can remove such steady bias. What is the quantitative advantage of the SMOS based method compared to such simple methods ? .

If the known/steady bias on rainfall was removed before assimilating soil moisture in the model, couldn't the complementarity between the high resolution rainfall information provided by the satellite products and the soil moisture information be better exploited ?

The improvement of the discharge simulation is very low in the case of TRMM based forcing (because the first order correction, i.e. strong bias removal, is not relevant in this case). In this case, what is the effect of moisture assimilation on other variables (ground water etc...) ; is the space/time distribution of water within the basin improved ? .

I believe this work would be more convincing if the post-adjusted version of TRMM 3B42v7 was also included and the questions above explored.

Note that bias corrected versions of PERSIANN (persiann-CDR) and Cmorph (v1) are also available for the study period and could be easily used in the present study for comparison.
* * *

---

## Referee Comment (RC2) · Anonymous Referee #2 · 1 Apr 2016

First of all, I would like to apologize for my very late review.

The study of Leroux et al. investigates the impact of satellite soil moisture data assimilation on simulated streamflow, soil moisture and water table depth in the Ouémé catchment in Benin. Using in situ measurements from this densely equipped river basin as a reference, the results demonstrate that the SMOS soil moisture data assimilation improves the simulations of all three variables regardless of the precipitation data product that is used to run the model. The results are in line with those obtained in a number of similar studies reported in the recent literature. The authors do not introduce any significant methodological developments, but rely on a rather standard data processing and assimilation procedure to do their analysis. This in itself is not a problem given

the fact that the number of such studies is still rather limited. Experiments with different soil moisture data sets, different models and in different experimental catchments are clearly welcome to get a better understanding of the advantages and limitations of satellite soil moisture data assimilation for hydrological predictions. The single most interesting result for me was the possibility to significantly improve streamflow and water table depth simulations of the model that uses in situ measured precipitation data as input data. This is the most challenging setup and it is an important result that improvements were obtained for such a scenario. It is more difficult to evaluate the meaning and merit of the results obtained when satellite rainfall products were used as forcing data. My main concern relates to the design of the related experiments. The hydrological model is first calibrated with in situ measured streamflow and soil moisture. The same optimal parameter set is then used when highly biased satellite precipitation data sets are used as forcing data. In fact the satellite soil moisture data is bias corrected before the assimilation is carried out. The reference seems to be the open loop of the calibrated models using in situ measured precipitation as input. I found this experimental setup questionable for two main reasons. First of all, it is rather obvious that the (unbiased) satellite soil moisture data assimilation will improve the (biased) soil moisture simulations that are obtained when using the different (biased) precipitation data sets as inputs. Second, I would argue that it is a very unrealistic scenario to assume that a model that has been calibrated with in situ measured discharge and soil moisture uses as input satellite precipitation products. The more likely scenario is that satellite precipitation is the only available data source for calibrating the model. Therefore, in my opinion, it would have been preferable to re-calibrate the model for each satellite precipitation product before carrying out the soil moisture data assimilation. The authors explain that they did not proceed like this because of the compensation effects that would impact the model parameters when biased forcing data are used. This is true, but in this experiment the compensation takes place whenever a soil moisture data set is assimilated. I have further some concerns regarding the methodology. It is not clear to me how the model error covariance matrix is defined. The authors mention

that fixed values are used but do not explain how this was done. Moreover, is seems as if the same values were used regardless of the input precipitation data. This would be a rather gross simplification as obviously the uncertainties of the state variables very much depend on the quality of the forcing data. The covariance matrix should reflect the fact that the satellite precipitation data are much less reliable than the in situ data. An approach based on the generation of ensemble members would be more adequate in my opinion. It is also not clear how the R matrix was setup. Is this just the variance obtained from each pixel's time series? I don't agree that this can be used to estimate the uncertainty of the observations. The analysis of the residuals with respect to the in situ data (where available) would provide a better estimate. Other than that there is triple collocation and its numerous variants. I found it also surprising that the variance attributed to the SMOS observations is that low. Finally, I noticed on Figure 6 that the SMOS observations are well distributed around the in situ measured soil moisture data but less so around the "open loop + precip in situ". This would suggest that a CDF matching was carried out with respect to the in situ data and not the open loop. Or are these the SMOS data points before a CDF matching was applied? In fact I am wondering how the SMOS data assimilation e.g. in May 2012 was able to lead to an almost perfect match with the in situ data when the "open loop + precip in situ" and the SMOS data points were both slightly biased in that time period.

---

## Editor Comment (EC1) · FF Fenicia (Editor) · 16 Apr 2016

I would like to thank the reviewers for their constructive comments. All reviewers suggested that major improvements to the paper are needed, and I invite the authors to carefully consider their comments.

The introduction starts too broad – in a hydrological journal it is not necessary to mention that water is important. The authors should go straight to the context of their work.

The introduction lacks a clear definition of the objectives. This may seem a minor omission, but judging the appropriateness of the methodology is very difficult if clear objectives are not formulated.

[Figure]

For example, the authors compare the performance of a model calibrated with a set of input, to the performance of the same calibrated model switching to another set of input. I agree with the point made by Reviewer 2, that in this case the model performance can only deteriorate. The other issue, is that the methodology is applicable only when ground stations are available. The question is why one would want to use satellite data if ground data are present. This question comes back to what are the real objectives of this work.

It seems to me that the states of the ground stations driven model are used as ground truth with respect to the satellite data driven model. I did not find a real justification for this approach, as both models can be wrong.

The hydrological model is called physically based, but then its parameters are calibrated. What is then the definition here used for physically based model? A physically based model is usually defined as one whose parameters should be directly observed.

The authors effectively combine results and discussion. I think these should be clearly separated.

The conclusion is a combination of summary and conclusion. Again, these should be separated.

---

## Author Comment (AC4) · 21 Apr 2016

Thank you very much for the fruitful comments you collected for this study.

We agree that the goal of this work has not been well defined at all. The two anonymous reviewers and Dr. Marielle Gosset gave us very good advice on how to improve the manuscript.

We are considering the remarks from all the reviewers and have been working hard on the reorganization of the manuscript. We would like to suggest the following layout for the submission (see below). Moreover, one author has joined us: Luc Séguis, HydroSciences, Montpellier, France. I hope it is possible to add him as co-author of

this paper when we will resubmit.

Thank you very much again for your work.

* Title

Assimilation of SMOS soil moisture into a distributed hydrological model and impacts on the water cycle variables over the Ouémé catchment in Benin

* Abstract

- quasi real time (RT) rainfall forcing are not accurate enough for hydrologic RT applications but post-adjusted rainfall products are only available several months after observations

- soil moisture assimilation of SMOS observations can correct for the inaccurate amount of water brought by RT rainfall forcing

- soil moisture is adjusted, which has a positive impact on water table depth and streamflow simulations, which can lead to a better management of available water resources and extreme events

*Introduction

- water cycle, hydrologic modeling, scarcity of in situ measurements in tropical regions, monsoon impact on people life

- RT rainfall forcing are not very good, but post-adjusted rainfall products are only available 2-3 months later

- assimilating soil moisture is assessed to correct for wrong amount of water brought by RT rainfall forcing

* Data, model and methodology

- in situ measurements (soil moisture, streamflow, precipitation)

- rainfall satellite products

— RT and post-adjusted and the delay of availability

— in situ and post-adjusted are very close

- DHSVM model

— description of the model

— calibration of the model using in situ or post-adjusted rainfall forcing

— simulations using RT rainfall forcing

- SMOS soil moisture

- assimilation method: optimal interpolation

— description of the method, the assumptions, how to deal with the resolution differences

— description of the experiment: simulations using RT rainfall forcing with SMOS soil moisture assimilation

* Results and discussion

- correction of the soil moisture (control variable)

- impact on the water table depth simulations

- impact on the streamflow simulations

*Conclusion

- soil moisture corrected by SMOS assimilation

- positive impact on the water table depth simulations => can lead to a better simulation and management of the actual ground water resources (RT application)
- positive impact on the streamflow simulations => can lead to a better simulation and management of extreme events such as floods during the monsoon period (RT application)

- this work shows the possibility to implement a near real time hydrologic framework for RT applications wherever it is possible to obtain a proper calibration of the hydrologic model beforehand, which is one limitation of this method (or optionally, the RT rainfall products could be directly corrected using SMOS observations, work in progress in LTHE, Grenoble)

- assimilation method needs to be improved and ensemble technics should be used to avoid any assumptions on the errors of the model and the observations

---

## Author Response (AR1)

Based on the comments from the two anonymous reviewers (AR#1 and AR#2), and from Dr. Marielle Gosset, we have decided to re-organize the whole manuscript and re-write most of the text in order to make the message clearer. The introduction is more focused, and the purpose of this study is now clearly defined. Some parts have been removed while others have been added (using reanalyzed precipitation products).

The objective of the study is to show the potential of implementing SMOS soil moisture assimilation using quasi-real time precipitation products compared to using only reanalyzed rainfall products that can take several months before being made available.

The new layout of the manuscript is as follows:
* * *
1- Introduction

2- Study area and satellite data
2-1- The Ouémé catchment and the in situ measurements
2-2- Satellite rainfall products
2-3- SMOS soil moisture product

3- Model and data assimilation
3-1- DHSVM model
3-2- Assimilation method: optimal interpolation
3-3- Statistics metrics

4- Results and discussion
4-1- Correction of the control variable: the soil moisture
4-2- Impact on the water table depth simulations
4-3- Impact on the streamflow simulations

5- Conclusions
* * *
We would like to thank all the reviewers for their comments and advice on how to improve the manuscript. We have already  answered the questions of the reviewers in the discussion but the copies of our answers are copied below. We strongly encourage and thank in advance the reviewers to read the new manuscript since most of it has been reviewed, especially the introduction (part 1) which introduced the objective of the study, the presentation of the rainfall product (real-time RT and reanalyzed RE, part 2.2), the presentation of the model with the difference of simulations between RT and RE (part 3.1), all the results and discussions (part 4) with the benefit of SMOS assimilation compared to RT rainfall alone and its performances compared to those with RE precipitations, and the conclusions (part 5). Most of the figures and tables have been updated with a comparison between RT alone, RT+SMOS assimilation, and RE alone.
* * *
Anonymous Referee #1

This study aims at assimilating SMOS soil moisture observations to correct for errors in precipiation and reduce uncertainties in simulated discharge for a catchment in Benin. The study uses 4 precipitation datasets to capture the uncertainties in the simulated soil moisture. This is a nice detailed study that goes beyond the standard work and should be considered for publications. However, I do feel that the authors can make some improvements and therefore recommend the revisions below. One more general remark is that the goals seems to be to improve streamflow simulations, while the paper reads more like a paper that does precipitation correction using satellite soil moisture and thereby obtain improved simulations.

*We would like to thank Anonymous Referee #1 for the constructive remarks, which made this work more thorough. Based on these remarks, the objective of the work has been refined in the abstract and hopefully throughout the whole manuscript.*
*In the original paper, the real-time rainfall products (PERSIANN, TRMM, and CMORPH) are used in the model, which bring too much water. Soil moisture assimilation can attenuate the effect of bringing too much water over the whole basin by correcting the soil moisture content. This positive bias in the real-time precipitation products has already been identified and corrected in reanalyzed datasets: PERSIANN-CDR, TRMM-v7, and CMORPH-v1. When these bias-corrected rainfall products are used, the simulations are much better and in very good agreement with in situ measurements (soil moisture, water table depth, streamflow). However, these products are only available at least 2 months after the real-time products. The objective of the paper is now focused on a proposition of a fair approach to fill the gap of these few months.*

Major remarks

Page 3, Line 8-13 Some recent studies have actually used assimilated SMOS soil moisture at a scale comparable to the catchment in the work (Lievens et al 2015 and Wanders et al 2014). These studies show that the assimilation of SMOS has a positive impact on the streamflow estimations in some scenarios. It is stated in the manuscript that it has no impact, which is contradicted by the above mentioned studies. I think a comparison with these studies would be valuable for the reader.

*The last part of the sentence has been removed ("but has little impact on the streamflow estimation") as it not true that all these studies found that SM assimilation did not improve the streamflow simulation. We apologize, it was a mistake. These references have been added in the introduction as proposed by the referee and as it is quite relevant for the reader as well.*

Some restructuring of the introduction would help to more clearly state the research gap that this paper would like to fill. I my opinion, the most novel thing done in this study is the use of multiple precipitation forcing product and the impact of SMOS DA on the hydrological simulations with these products. Now it states (Page 4, line 16-17) that SMOS assimilation impact on streamflow is the main goal, while in Page, Lines 13 it was stated that assimilation of SMOS has no impact on the streamflow performance.

*Some parts of the introduction have restructured and reorganized. The last two paragraphs of the introductions have been modified so that the main objective of this work is now clearly stated:*
*"Reanalyzed versions of the satellite precipitation products, correcting for their initial inaccuracies, are often available but only after several weeks or months after the observations,*

*which can be an issue for operational systems."*

*"The objective of this study is to constrain the water and energy balances by assimilating surface soil moisture satellite observations using the near-real time satellite rainfall products.Our study focuses on the assimilation of SMOS soil moisture over a West African catchment in Benin and investigates its impact on other hydrological variables. A first part of this article presents the Ouémé catchment, the in situ measurements and the satellite data. Then the hydrological model DHSVM is briefly described along with the assimilation method. The results of the assimilation are presented in the last section before the conclusions."*

Page 5 Line 25-26 Using rainfall satellite product doesn't make it challenger. If one would use perfect rainfall data the potential of SMOS for streamflow improvements would be almost zero, while if the rainfall is very imperfect the potential impact is significant since the initial guess is far of and the potential improvement is large. Please remove or correct this incorrect statement.

*The referee is absolutely right. This sentence is very confusing as the further you start from the truth, the bigger impact the assimilation will have. So this case should not be called "challenging" but the exact opposite since it is expected to get the best improvement. The whole sentence has been removed.*

Page 5 Line 31, Why is the 3B42RT product used instead of the reanalysis product of TRMM, which is gauge corrected and therefore has a higher quality compared to reality.

*As other referees suggest, the reanalyzed rainfall products have been added to the study in order to compare their performances with those of the assimilation. And, after bias-correction, the open-loop give much better results than the real-time products.*

Table 2, how is it possible that the quality of the SM simulations after assimilation show a decreased performance compared to before assimilation. Does this mean that SMOS and the observations are not well aligned or is the DA procedure sub-optimal? Does result is at least counter intuitive to what one would expected after DA of additional observations.

*As explained in the text, only the bias results should be judged. Correlation, sdd and rmse are impacted by the discontinuities introduced by the assimilation when the soil moisture corrections are applied. If the observation is drier than the simulation, then water is removed from the ground and a discontinuity appears, which artificially increases the sdd and the rmse and lowers the correlation.*
*In the case of in situ precipitation, the assimilation does not improve the performances as it tends to only add noise to simulations that are already good. In the cases of real-time precipitation products, the bias are always reduced after assimilation.*
*However, the assimilation technique implemented here is quite basic since it is the Optimal Interpolation. With this technique, the B and R matrix are set by the user and do not evolve in time. So the DA might not be as optimal as an Ensemble Kalman Filter could be.*

I think some maps of the spatial improvement of the simulations would help the reader to get a better feeling with regard to where the largest potential is for further improvement. Is it the upstream areas or are better results obtained in other regions.

*In order to draw maps of improvement of surface parameters such as the soil moisture, a map of the*

*"true" state is necessary. In situ measurements are only available at several locations which are not enough to interpolate for the whole basin.*
*Regarding the streamflow simulations, only two points at the outlet of each sub-basin are simulated, so it is difficult to say if the improvements/changes are better upstream or downstream, but it would be interesting to simulate a streamflow at each point of the river and compare them with available in situ measurements.*

Minor remarks

With respect to the precipitation corrections that are in a way done I think it would be

useful to mention here some studies that focus on this aspect (e.g. Crow & Bolten,

2007; Crow et al. 2011; Pellarin et al 2013; Wanders et al 2015)

*Thank you very much for these references. In the introduction, a paragraph has been added explaining that several studies have showed that it was possible to correct real-time precipitation products using satellite observations. And it has been added that the present study is different in the sense that the correction of these inaccurate real-time precipitations is operated within the hydrological model, as opposed to the references given here.*

Figure 1, the quality of the image low in my version of the manuscript

*It was not supposed to be a low quality image. It is not the case in the author version. But I will check on the revised version.*

Page2, Line 14 in space should be spatial

*It has been corrected, thank you.*

Page 2, Line 21, 0.04 is only the mission requirement of SMOS and not its actual

accuracy or provide a reference to confirm this.

*The reviewer is correct, this is a target accuracy set by the mission requirements. Modifications have been added to the text ("with a mission requirement accuracy of 0.04 m3/m3").*

Table 3 This table tells me with far from perfect precipitation one can gain a lot from

the assimilation of satellite derived SM data, while if the forcing is almost perfect the

assimilation of SM is a difficult and potentially low gain approach. Maybe some of this

should be mentioned in the dicussion

*Some words have been added to the paragraph beginning with "Another representation of these statistics is the Taylor diagram in Fig....".*

Table 4 The % are not well explained in the caption of the table, please adjust. TRMM

after assimilation, should that not be -6%?

*The % are actually explained at the bottom of the table ("statistics compared to OL simulations using in situ precipitations"), but this description should be mentioned in the caption instead. It has been moved and changed to "The percentages between parenthesis indicate the comparison with the OL simulations using the in situ precipitations that are used for reference.". Thank you for the suggestion.*

Anonymous Referee #2

The study of Leroux et al. investigates the impact of satellite soil moisture data assimilation on simulated streamflow, soil moisture and water table depth in the Ouémé catchment in Benin. Using in situ measurements from this densely equipped river basin as a reference, the results demonstrate that the SMOS soil moisture data assimilation improves the simulations of all three variables regardless of the precipitation data product that is used to run the model. The results are in line with those obtained in a number of similar studies reported in the recent literature. The authors do not introduce any significant methodological developments, but rely on a rather standard data processing and assimilation procedure to do their analysis. This in itself is not a problem given the fact that the number of such studies is still rather limited. Experiments with different soil moisture data sets, different models and in different experimental catchments are clearly welcome to get a better understanding of the advantages and limitations of satellite soil moisture data assimilation for hydrological predictions. The single most interesting result for me was the possibility to significantly improve streamflow and water table depth simulations of the model that uses in situ measured precipitation data as input data. This is the most challenging setup and it is an important result that improvements were obtained for such a scenario. It is more difficult to evaluate the meaning and merit of the results obtained when satellite rainfall products were used as forcing data. My main concern relates to the design of the related experiments. The hydrological model is first calibrated with in situ measured streamflow and soil moisture. The same optimal parameter set is then used when highly biased satellite precipitation data sets are used as forcing data. In fact the satellite soil moisture data is bias corrected before the assimilation is carried out. The reference seems to be the open loop of the calibrated models using in situ measured precipitation as input. I found this experimental setup questionable for two main reasons. First of all, it is rather obvious that the (unbiased) satellite soil moisture data assimilation will improve the (biased) soil moisture simulations that are obtained when using the different (biased) precipitation data sets as inputs. Second, I would argue that it is a very unrealistic scenario to assume that a model that has been calibrated with in situ measured discharge and soil moisture uses as input satellite precipitation products. The more likely scenario is that satellite precipitation is the only available data source for calibrating the model. Therefore, in my opinion, it would have been preferable to re-calibrate the model for each satellite precipitation product before carrying out the soil moisture data assimilation. The authors explain that they did not proceed like this because of the compensation effects that would impact the model parameters when biased forcing data are used. This is true, but in this experiment the compensation takes place whenever a soil moisture data set is assimilated. I have further some concerns regarding the methodology. It is not clear to me how the model error covariance matrix is defined. The authors mention that fixed values are used but do not explain how this was done. Moreover, is seems as if the same values were used regardless of the input precipitation data. This would be a rather gross simplification as obviously the uncertainties of the state variables very much depend on the quality of the forcing data. The covariance matrix should reflect the fact that the satellite precipitation data are much less reliable than the in situ data. An approach based on the generation of ensemble members would be more adequate in my opinion. It is also not clear how the R matrix was setup. Is this just the variance obtained from each pixel's time series? I don't agree that this can be used to estimate the uncertainty of the observations. The analysis of the residuals with respect to the in situ data (where available) would provide a better estimate. Other than that there is

triple collocation and its numerous variants. I found it also surprising that the variance attributed to the SMOS observations is that low. Finally, I noticed on Figure 6 that the SMOS observations are well distributed around the in situ measured soil moisture data but less so around the "open loop + precip in situ". This would suggest that a CDF matching was carried out with respect to the in situ data and not the open loop. Or are these the SMOS data points before a CDF matching was applied? In fact I am wondering how the SMOS data assimilation e.g. in May 2012 was able to lead to an almost perfect match with the in situ data when the "open loop + precip in situ" and the SMOS data points were both slightly biased in that time period.

*We would like first to thank anonymous reviewer #2 (AR#2) for her/his comments and remarks. Some changes have already been made according to comments from other reviewers. The focus of the study was not well defined and some of the reviewers already highlighted the lack of purpose. We have worked on that aspect and put this study in an operational context with possible real-time applications.*

*As AR#2 mentioned, the calibration of the model is realized using in situ precipitations, which is then used with satellite precipitations. This can be confusing because if the in situ precipitations are available, why would someone use the satellite products. In our case, (and it has not really been mentioned in the text), the in situ precipitations have been produced for this study using Lagrangian kriging at very high resolution (Vischel et al., 2011). This product is not an operational product and is generated on demand. The reprocessed satellite precipitation products (post-adjusted observations using in situ networks) are only available several weeks after the observations and cannot be used operationally. These post-adjusted satellite precipitation products are very close to the in situ measurements. Using either product would lead to the same calibration. Here, in an operational context, only real-time satellite precipitations are available and these products should not be used for calibration (as advised in Bitew et al., 2011) as they are not accurate enough: not enough or too much water, and often not at the right time of the year.*
*What has been done here and what makes more sense to us, is to use the in situ precipitations generated for 2010 for calibration and use this calibration for the following years. This calibration is supposed to be good if the right amount of water is given in. Of course, this is not the case when the real-time satellite precipitations are used and by assimilating SMOS soil moisture products in the model, we hope to achieve that. By assimilating SMOS soil moisture products, we go back to the case where the appropriate amount of water is in the model, and the previous calibration, after assimilation, is then appropriate.*

*The main purpose of the study is to be able to run the model operationally in quasi real time (a few days delay) using real time precipitation forcing and soil moisture assimilation.*

*In order to illustrate that, simulations using post-adjusted satellite products have been added to the manuscript showing that they perform as good as the in situ precipitations. After assimilation, the performances are almost as good (similar using near real-time PERSIANN+SMOS, a bit lower for the other two).*

*Regarding the assimilation methodology, we agree that the one used here is a rather gross simplification of what should be ideally done. By choosing an optimal interpolation, implementation simplicity has been preferred as it was our first attempt of data assimilation using this model. Ensemble methods need computer resources and skills that were not available at the time of this study. With the experience of this first exercise, it would make no doubt that ensemble or extended Kalman filtering methods would provide much more reliable results. However, the question of the optimal perturbation to be applied for the ensemble generation or for the Jacobian matrix is still an open question to the data assimilation community, and is for now up to the user.*

*As a first attempt of R and B estimations, the variances and the covariance of the variables have been used. We totally agree with AR#2, this is not the correct way to compute the errors. However, we were more looking at the ratio between the model and the observations and were looking for "fair" trade between these two quantities, i.e. K around 0.5 for the assimilation layer giving as much weight to the model and to the observations. This is the case here, for the 2nd soil layer, where B is 0.019 m3/m3 and R is 0.017 m3/m3. We are aware this is a huge simplification, and so is the assumption that R is diagonal. With a large satellite footprint, this is not true to assume that there is no covariance between the observations. In general we agree that big improvements can be realized by using more sophisticated methods in the assimilation process. Here, we just implemented everything from scratch and wanted to see the impact of a surface variable (soil moisture) on a more integrated variable (such as the streamflow at the outlet). In future developments, ensemble Kalman filter will be used to avoid this kind of assumptions.*

*Regarding the CDF matching, it has been done using open-loop simulations and not the in situ soil moisture measurements. The reason why SMOS appears closer to in situ on Figure 6 is only a coincidence in this case. CDF matching is actually performed at SMOS scale, i.e. at 25 km, which requires to upscale the model simulation from its finer resolution (1 km) to SMOS resolution. In order to achieve the upscale, a simple average of the model pixels contained in the SMOS pixel is performed.*
*However, the adjustment of the soil moisture is realized at the model scale. So, for each model pixel, all the SMOS observations contained in a so called 'influence radius' (set here at 40 km, corresponding to the SMOS instrument resolution) and weighted according to their distance to the considered model pixel. On Figure 6, the SMOS yellow squares represent the closest SMOS observations but not the assimilated soil moisture.*
*This methodology is the same as the one defined in deLannoy et al., 2010 (called "3DCm").*

*\*\*\*\*\*\**

*\* Bitew M.M., and Gebremichael M.. Assessment of satellite rainfall products for streamflow simulation in medium watersheds of the Ethiopian highlands. Hydrology and Earth Science Systems, vol. 15, pp. 1147-1155, 2011.*
*\* De Lannoy G., Reichle, R., Houser, P., Arsenault, K., Verhoest, N., and Pauwels, V.. Satellite-scale snow water equivalent assimilation into a high-resolution land surface model, Journal of Hydrometeorology, vol. 11, pp. 352–369, 2010.*
*\* Vischel T., Quantin G., Lebel T., Viarre J., Gosset M., Cazenave F., and Panthou G.. Generation of high-resolution rain fields in West Africa: evaluation of dynamic interpolation methods. Journal of Hydrometeorology, vol. 12, pp. 1465-1482, 2011.*
*(this last reference will be added to the manuscript bibliography)*

*\*\*\*\*\*\**

Dr. Marielle Gosset's short comment

This is an interesting work, illustrating how satellite observation of rainfall and soil moisture can be complementary. However the objective of the work should be better explained and the choice of using only non-adjusted (or RT) satellite rainfall products should be better justified. If the objective of this work is to propose an alternative bias-correction method for RT satellite rainfall than the operationnal advantage of the proposed method should be developped. If the objective is to show how combining information on both rainfall and soil moisture can help a better understanding/modelling of hydrological processes, than the point would be strenghened by adding post-adjusted products (such as 3B42V7) in the study.

*Thank you very much for your comment. Your remarks helped the authors to put this work in a different perspective. The operational point of view of this method is actually quite interesting and has been added to the study. The performances of the real-time precipitations, the assimilation of SMOS using the real-time precipitations, and the re-analyzed precipitations are compared. More specifically and as suggested, the following post-adjusted precipitations products have been used: PERSIANN-CDR, TRMM-v7 3B42, and CMORPH-v1 CRT. The study shows that SMOS assimilation can perform as good as the post-adjusted precipitation products (especially for PERSIANN, and a bit less for the other two products).*

-is the SMOS based bias correction potentially available with better delay than what is currently done based on gauges (for instance to correct 3B42RT into 3B42v7 ) ? what are the current/future perspective on soil moisture monitoring and would the expected sampling allow for using soil moisture based bias correction to be used operationnally?

*The version of the SMOS product that is used here is the Level 3 soil moisture, usually available under 8 days after the observations. SMOS observed brightness temperatures are made available in quasi real time (2 to 3 hours), which can allow an assimilation of the observations in quasi real time (cf. work of Nemesio J. Rodriguez-Fernandez, using neural network to retrieve soil moisture within one day). However, the post-adjusted precipitation products are only available weeks after the actual observation (up to a couple of months). SMOS would definitely be useful during the release of the real-time precipitation and the adjusted products.*

-One of the tested product (PERSIANN) has been shown by many previous authors (cited in the present paper) to have a large and steady positive bias over the region. Simple method (like pdf matching based on gauges series used by Thiemig et al, among others) can remove such steady bias. What is the quantitative advantage of the SMOS based method compared to such simple methods ?

*By assimilating SMOS soil moisture, a physical information is added into the process regarding the amount of soil moisture and also its spatial distribution (even at a coarse scale). A pdf matching is a statistical method that consists of matching certain statistics (such as the mean and the standard deviation) to those of in situ gauges. This statistic method can be extremely efficient and very fast if in situ gauges are available indeed, just like they are incorporated in the post-adjusted precipitation products. Since they are not always available everywhere, as opposed to satellite observations, SMOS has the advantage to adjust the soil moisture quantity in regions where there is no other precipitation network.*

If the known/steady bias on rainfall was removed before assimilating soil moisture in the

model, couldn't the complementarity between the high resolution rainfall information provided by the satellite products and the soil moisture information be better exploited?

*It appears here that the bias is not steady since there can be precipitations seen in the real-time products during the dry season (spring mainly), whereas some of the rainfall events during the rainy season can be underestimated. Moreover, the bias or errors in the satellite precipitation products can only be identified if rain gauges are present on site, which is not always the case. The complementarity of the high resolution of the rainfall products and the soil moisture information is not properly addressed here, but it is one of the current subject under study at LTHE, Grenoble.*

The improvement of the discharge simulation is very low in the case of TRMM based forcing (because the first order correction, i.e. strong bias removal, is not relevant in this case). In this case, what is the effect of moisture assimilation on other variables (ground water etc...) ; is the space/time distribution of water within the basin improved?

*Regarding the impact of the assimilation on the groundwater simulations, they are shown on Figure 10 of the manuscript. They are all improved by the assimilation compared to the in situ measurements of groundwater (at one location). Unfortunately, there are not enough in situ locations to assess the quality of the spatial distribution of the water within the basin.*

I believe this work would be more convincing if the post-adjusted version of TRMM 3B42v7 was also included and the questions above explored.
Note that bias corrected versions of PERSIANN (persiann-CDR) and Cmorph (v1) are also available for the study period and could be easily used in the present study for comparison.

*The three post-adjusted versions of the precipitation products have been added to the manuscript. It allows this work to be more thorough and focused on the real advantage of SMOS. We would like to thank Marielle Gosset for her valuable comments which made this work more focused.*